# Secretory granule protein chromogranin B (CHGB) forms an anion channel in membranes

Gaya P Yadav[1], Hui Zheng[2], Qing Yang[3], Lauren G Douma[4], Linda B Bloom[4], Qiu-Xing Jiang[1]

**Regulated secretion is an intracellular pathway that is highly conserved from protists to humans. Granin family proteins were proposed to participate in the biogenesis, maturation and release of secretory granules in this pathway. However, the exact molecular mechanisms underlying the intracellular functions of the granin family proteins remain unclear. Here, we show that chromogranin B (CHGB), a secretory granule protein, inserts itself into membrane and forms a chloride-conducting channel. CHGB interacts strongly with phospholipid membranes through two amphipathic α helices. At a high local concentration, CHGB insertion in membrane causes significant bilayer remodeling, producing protein-coated nanoparticles and nanotubes. Fast kinetics and high cooperativity for anion efflux from CHGB vesicles suggest that CHGB tetramerizes to form a functional channel with a single-channel conductance of ~125 pS (150/150 mM Cl⁻). The CHGB channel is sensitive to an anion channel blocker and exhibits higher anion selectivity than the other six known families of Cl⁻ channels. Our data suggest that the CHGB subfamily of granin proteins forms a new family of organelle chloride channels.**

## Introduction

Cells rely on secretory pathways to send out specific bioactive molecules to their surroundings (Grimes & Kelly, 1992). Both constitutive and regulated secretory pathways have been studied extensively since early EM observations of intracellular vesicular trafficking. Many molecular players in these pathways are known. Conserved mechanisms for vesicle targeting and membrane fusion are established (Farquhar & Wellings, 1957; Jamieson & Palade, 1964; Rothman, 1994; Salama & Schekman, 1995; Rizo & Xu, 2015). The three key steps of regulated secretion are granule biogenesis at the TGN, granule maturation through fusion, sorting and membrane shedding, and exocytotic release of mature granules (Hosaka & Watanabe, 2010). Maturation of nascent immature secretory granules (ISGs) into dense-core granules (DCGs) is accompanied by H⁺

-ATPase–driven acidification and resorting of granule proteins and cargos (Johnson & Scarpa, 1976; Johnson et al, 1982).

Granin family proteins are by default granule proteins that chaperone other secretory molecules through regulated secretion (Hosaka & Watanabe, 2010; Bartolomucci et al, 2011). They have both intracellular and extracellular functions. Chromogranins (CHGs) are believed to function in all three steps of the regulated secretory pathway. It was proposed that CHGs interact with cargos and serve as a low-affinity, high-capacity Ca²⁺ reserve. Their extracellular functions are executed by CHG-derived peptides that are associated with various human diseases (Bartolomucci et al, 2011). However, the molecular mechanisms for all CHGs' intracellular functions remain to be elucidated.

Members of the granin superfamily (Bartolomucci et al, 2011) share low amino acid sequence similarity (Montero-Hadjadje et al, 2008) and are clustered into phylogenetic subfamilies. Granin proteins of distinct subfamilies usually coexist in secretory granules and were hypothesized to work with different partners (Bartolomucci et al, 2011). Native chromogranin B (CHGB) forms high-order aggregates at low pH and with milliMolar Ca²⁺ (Yoo, 1995a, 1995b). It is genetically associated with type 2 diabetes, neurodegeneration, and psychiatric disorders (Gros-Louis et al, 2009; Davenport et al, 2010; Dominguez et al, 2012; Fuchsberger et al, 2016). It was proposed to participate in forming TGN proteinaceous aggregates and driving granule biogenesis in certain cell types (Tooze, 1998; Takeuchi & Hosaka, 2008; Diaz-Vera et al, 2012). Studies of CHGB–*null* mice by two groups reported cell-specific variations regarding CHGB's role in granule biogenesis, but similar roles in cargo maturation or monoamine loading into granules (Zhang et al, 2009; Diaz-Vera et al, 2010; Obermuller et al, 2010; Zhang et al, 2014). Even though CHGB has been prepared in heat-stable soluble fractions as protein complexes, partially purified native CHGB interacted quite strongly with lipid vesicles (Yoo, 1995b). A "tightly membrane-associated form" of CHGB was observed on the surface of PC-12 cells after exocytotic release of secretory granules (Pimplikar & Huttner, 1992). CHGB thus has dual states—outside or inside membranes. However, a mechanistic view on CHGB's functions in these two states is still unavailable.

[1]Department of Microbiology and Cell Science, University of Florida, Gainesville, FL, USA    [2]Department of Physiology, University of Texas Southwestern Medical Center at Dallas, Dallas, TX, USA    [3]Crop Designing Center, Henan Academy of Agricultural Sciences, Zhengzhou, PR China    [4]Department of Biochemistry and Molecular Biology, University of Florida, Gainesville, FL, USA

Correspondence: qxjiang@ufl.edu

In this work, we investigate the functional properties of CHGB proteins in different membrane systems and demonstrate that CHGB inserts into membrane and by itself suffices to form an unconventional chloride channel in vitro. A companion article will study the CHGB channel function in neuroendocrine cells.

# Results

### CHGB inserts into membranes and induces nanoparticles and nanotubules from bilayers

To study CHGB function, recombinant murine CHGB was purified from *sf9* cells (Fig S1A). During biochemical preparation, Triton X-100–like detergents were needed to keep CHGB soluble. In size-exclusion chromatography (SEC), purified CHGB was eluted as a single, symmetric peak (Fig 1A) with an apparent size equivalent to an ~300 kD globular protein. Because of posttranslational modifications, a high content of charged residues, or possibly detergent-binding, the recombinant CHGB ran at ~86 kD in a reducing SDS–PAGE gel (Fig 1B and C), which is larger than the 78 kD calculated from its sequence, behaving similarly to mature human CHGB (Pimplikar & Huttner, 1992). Because of the detergent micelle (~100 kD), the CHGB in detergents are likely a dimer, instead of a trimer. The detergent-solubilized CHGB treated with a bifunctional cross-linker, 4-(N-maleimidomethyl) cyclohexane-1-carboxylic acid 3-sulfo-N-hydroxysuccinimide ester (sulfo-SMCC), showed cross-linked dimers, trimers, tetramers, and high-order oligomers ($U_2$, $U_3$, $U_4$, and $U_n$ in Fig 1B), indicating a dynamic equilibrium between dimers and oligomers ($U_n$, n ≥ 3) and the dominance of dimers in detergents. Consistently, small amounts of CHGB oligomers were observed in an earlier SEC step during purification (Fig S2A).

Calcium binding is a biochemical hallmark of CHGB because of a high content of negatively charged residues. We examined calcium-induced CHGB aggregates by light-scattering and negative-stain EM (Fig S2B–F). Fitting of light-scattering data with a Hill equation yielded an apparent $k_D$ ~ 0.25 mM for $Ca^{2+}$, and a Hill coefficient of ~2.0 (Fig S2B), agreeing with CHGB's low affinity and high capacity for $Ca^{2+}$ (Yoo & Lewis, 1996). EM examination found that without $Ca^{2+}$ CHGB in detergents was monodispersed (Fig S2C). The size and amount of $Ca^{2+}$-induced aggregates increased significantly when $[Ca^{2+}]$ rose from 0.2 to 5.0 mM (Fig S2D–F).

To resolve the ambiguity between CHGB dimers and trimers in detergents, we performed single particle reconstruction. More than 5,400 particle images of negatively stained CHGB molecules (Fig S2G) were assembled for multi-variate statistical analysis (MSA). The eigenimages from MSA (top in Fig 1C) showed strong C2 symmetry. Angular reconstitution and 3D refinement yielded a negative-stain map at ~30 Å (Fig S2H). CHGB's small mass made it difficult to visualize under cryoEM. We used a ChemiC (chemically functionalized carbon) method (Llaguno et al, 2014) to enhance cryoEM image contrast (Fig S2I). More than 24,000 cryoEM images of CHGB molecules were manually selected for analysis. Multi-rounds of 2D and 3D classification led to a homogenous set of ~6,900 particles and a 3D map at ~10 Å resolution (Fig 1D), estimated by the gold-standard Fourier shell correlation (0.143). The cryoEM map has C2 symmetry and an elongated shape, explaining its larger apparent size in SEC (Fig 1A). Its distinct features may correspond to the predicted helical and random-coil regions, and verify CHGB's dimeric nature in detergents (Fig 1A and C).

The surprising need of detergents for CHGB stability in solution intrigued us to study whether it directly interacts with membranes using a vesicle flotation assay (Zheng et al, 2011). After reconstitution with phospholipids (Lee et al, 2013), CHGB floated with vesicles from bottom to top in a three-step Ficoll 400 gradient (arrow in the top of Fig 1E). Quantification of the protein bands revealed that >98% of CHGB protein was in membrane (Fig 1E, bottom panel), suggesting that with sufficient membrane surfaces, CHGB prefers to be in the membrane-associated state.

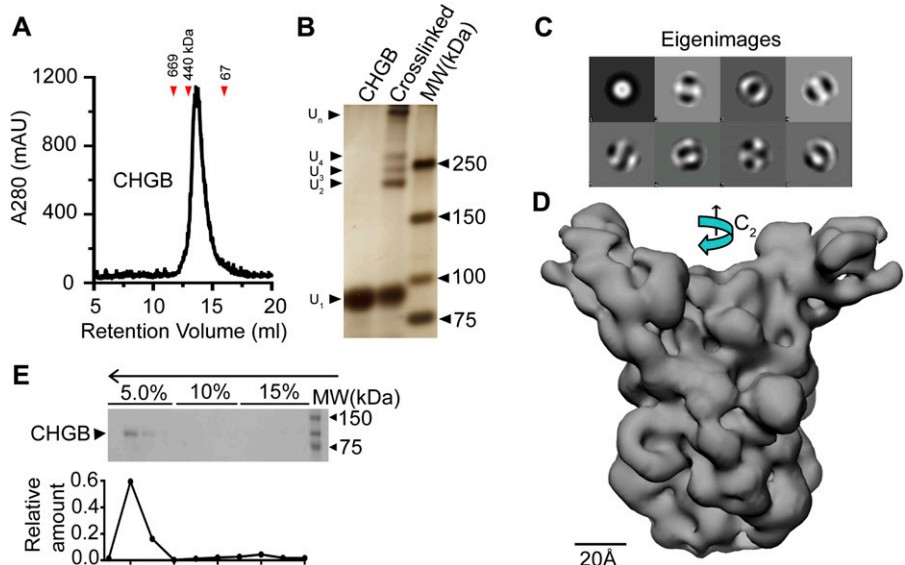

**Figure 1. Strong interaction of CHGB and lipid membrane.**
**(A)** SEC profile for purified CHGB. Red triangles indicate molecular weight markers. **(B)** Silver-stained SDS–PAGE gel of dimers ($U_2$), trimers ($U_3$), tetramers ($U_4$), and high-order oligomers ($U_n$) in CHGB treated with sulfo-SMCC (middle). **(C)** First eight eigenimages from MSA of a negative-stain EM dataset show C2 symmetry. **(D)** A cryoEM map for CHGB dimer at ~10 Å. **(E)** Top: SDS–PAGE of fractions from a vesicle floatation assay of CHGB in egg PC vesicles in a three-step Ficoll 400 gradient. The top arrow marks the vesicle floatation from bottom (15%) to top (5%). Bottom: quantified CHGB distribution from SDS–PAGE. The floatation experiments were repeated more than 20 times for different batches of CHGB vesicles.

CHGB may cause significant membrane remodeling. To visualize the effects of CHGB-membrane interaction on the bilayer structure, we examined CHGB vesicles by negative-stain EM (Fig 2A–D). When CHGB:lipid molar ratio (PLR) ≥ 1:1,000, equivalent to ~40 CHGB dimers per 100-nm vesicle, 25-nm nanospheres appeared on vesicles (red arrowheads in Fig 2B). More CHGB led to ~20-nm-thick nanotubules capped with 25-nm-diameter hemi-nanospheres (Fig 2C and D). In some specimens, the nanospheres were severed into individual soluble nanoparticles (Fig 2C). These results reveal CHGB-induced remodeling of membranes when local PLR is high, a condition likely being satisfied at the TGN sites for granule biogenesis, which might contribute to the heat-stable or soluble fractions used in previously published studies (Benedum et al, 1986; Benedum et al, 1987). The nanotubules and nanospheres indicate strong positive curvature caused by CHGB. To avoid membrane remodeling, we intentionally limited the PLR < 1:5,000 in most vesicle-based assays.

Lipid membranes drive CHGB oligomerization. When purified, CHGB was first reconstituted into vesicles and then extracted with detergents plus ~0.1 mg/ml lipids, >70% of CHGB protein was eluted by SEC at a position equivalent to an ~0.8 MD globular protein, suggesting that lipids may stabilize a high-order oligomer. Because of the elongated shape of the cryoEM map (Fig 1D), biochemically cross-linked tetramers (Fig 1C) and the detergent micelles, the smallest lipid-stabilized oligomers may be a tetramer ($U_4$; Fig S2A). It may endow the function of CHGB in membrane.

CHGB is inserted in membrane. When CHGB vesicles were treated with trypsin, almost all full-length CHGB (Fig 2E) was digested after ~50 min. With $Ca^{2+}$-loaded CHGB vesicles (Fig S3A) and a $Ca^{2+}$-sensitive dye, Indo-1, in the outside, we discovered that trypsin treatment did not break the membranes to cause any leak of $Ca^{2+}$, meaning that trypsin did not penetrate into the interior of the vesicles (Fig S3D). The nearly complete digestion of CHGB in vesicles indicates that almost all full-length CHGB proteins were inserted into membranes from the outside of the vesicles. A stable fragment out of trypsinization was membrane-protected and was named the membrane-interacting fragment (MIF) of CHGB (Fig 2E). The asymmetric membrane insertion of CHGB is expected to introduce positive curvature into membranes (Fig 2F) and lead to the nanotubules and nanoparticles (Fig 2B–D).

## Amphipathic segments are responsible for strong CHGB-membrane interaction

To examine the biophysical nature of CHGB-induced membrane curvature, next, we tested whether CHGB causes membrane breakdown. A fluorescein-release assay was implemented (Mukherjee et al, 2014) to monitor the increase in fluorescence when the carboxy fluorescein was released and unquenched. For this assay, valinomycin, a $K^+$ carrier, in DMSO (dimethyl sulfoxide) was introduced to eliminate charge accumulation. Our data showed that CHGB vesicles did not leak the 10-Å fluorophore (Fig 3A). When CHGB vesicles were loaded with $Ca^{2+}$, Indo-1 in the outside detected no leak of $Ca^{2+}$ (Figs 3B and S3A). As a positive control, purified type 1 inositol 3,4,5-trisphosphate ($IP_3$) receptors ($IP_3$R1) in vesicles generated $IP_3$-triggered $Ca^{2+}$ release (Fig S3B). The CHGB vesicles are thus very tight and the CHGB does not conduct $Ca^{2+}$.

We then investigated if CHGB vesicles release $Cl^-$ by using a Ag/AgCl (silver/silver chloride) electrode to measure $Cl^-$ as described by Stockbridge et al (2013). CHGB vesicles containing 300 mM KCl were changed into a buffer with 300 mM K-isethionate plus 0.2 mM KCl before being added into a recording chamber (diagramed in Fig S3C). Adding valinomycin initiated a $Cl^-$ signal that was a result of anion release from vesicles (bottom trace in Fig 3C). Control vesicles without CHGB showed no signal (top trace in Fig 3C). These findings suggest that a significant fraction of CHGB vesicles released their

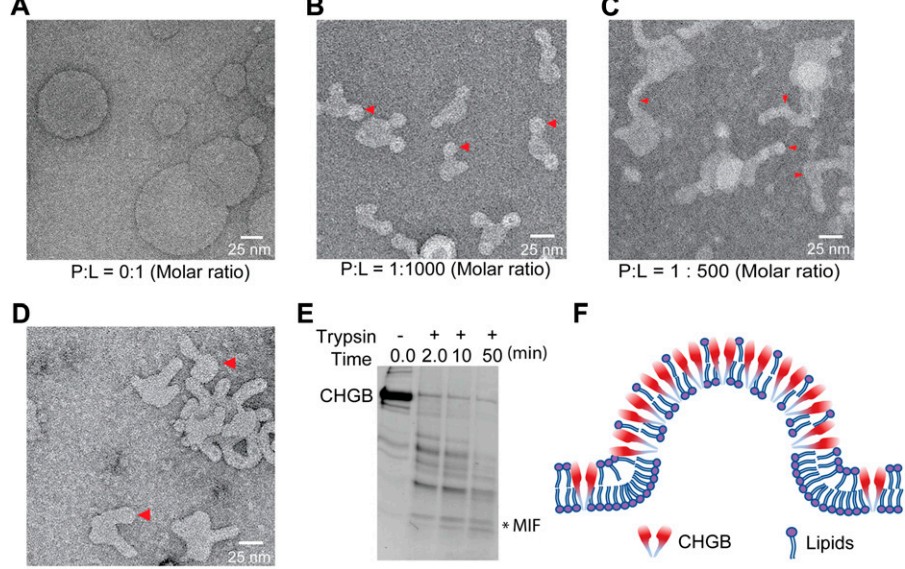

**Figure 2. Membrane insertion of CHGB introduces positive curvature in bilayers.**
Purified CHGB protein was reconstituted in lipid vesicles made of egg PC. Empty vesicles were prepared in the same way without CHGB. The vesicles were diluted by fivefold in a buffer containing 20 mM Tris–HCl, pH 7.4, 100 mM NaCl, 1.0 mM EDTA, and 2.0 mM β-ME before being loaded on carbon-coated grids and stained with 2.0% PTA (phosphotungstic acid/KOH, pH 8.0). Typical images of empty vesicles (**A**), CHGB reconstituted vesicles at 1:10 protein/lipid weight ratio (1:1,000 in molar PLR or ~40 CHGB dimers per 100 nm vesicle) (**B**), CHGB reconstituted vesicles at 1:5 protein/lipid weight ratio (1:500 in molar PLR or ~80 CHGB dimers per 100 nm vesicle) (**C**), and CHGB-reconstituted vesicles at 1:1 protein/lipid weight ratio (1:100 in molar PLR) (**D**) are shown. The experiments were repeated more than three times with very similar results. Budding compartments with positive curvature are marked with red arrowheads in (B–D). In (**B**), the nanospheres are marked at the tops of the nanotubules. Scale bars in A–D are all 25 nm. **(E)** 10 μg CHGB in egg PC vesicles treated with trypsin at RT. PMSF was used to stop the reaction. Samples collected at various time points were separated in a 4–20% SDS–PAGE gel. The stable short fragment at ~20 kD was collected for mass spectrometry analysis (marked as MIF), which is named as CHGB-MIF in Fig 3D–F. **(F)** Model for insertion of CHGB in membranes to cause positive curvature.

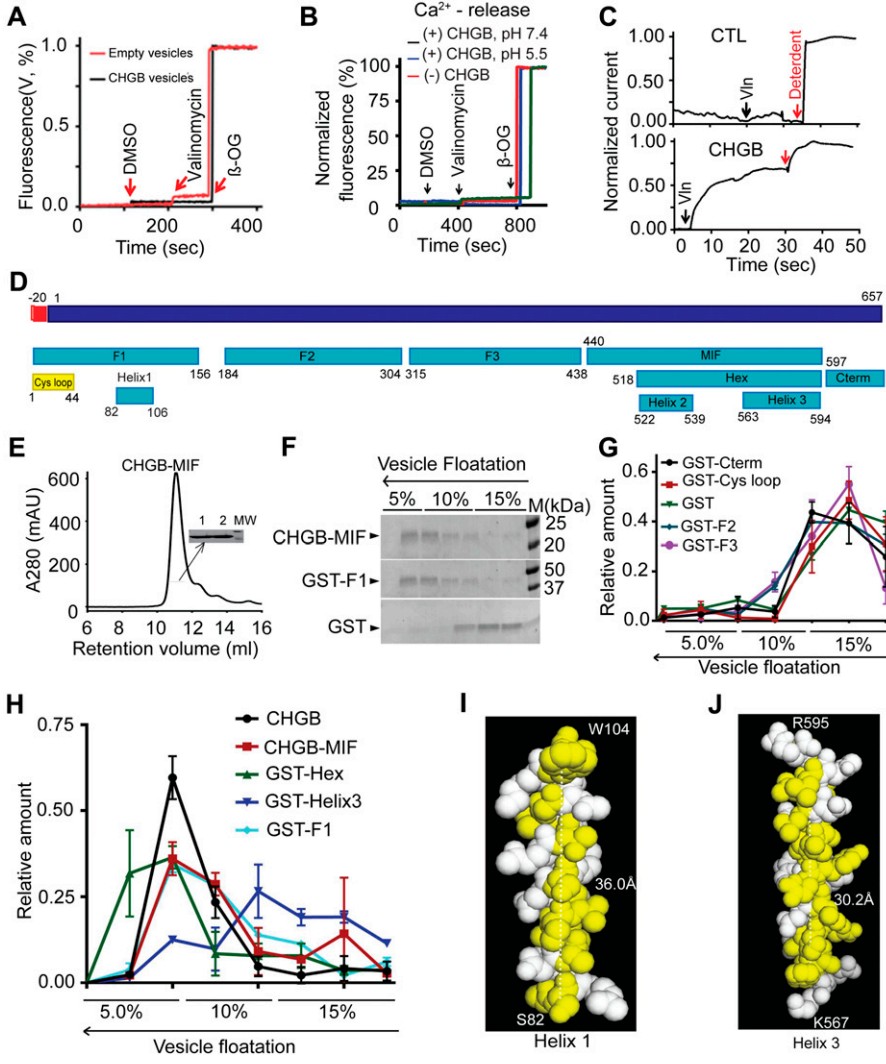

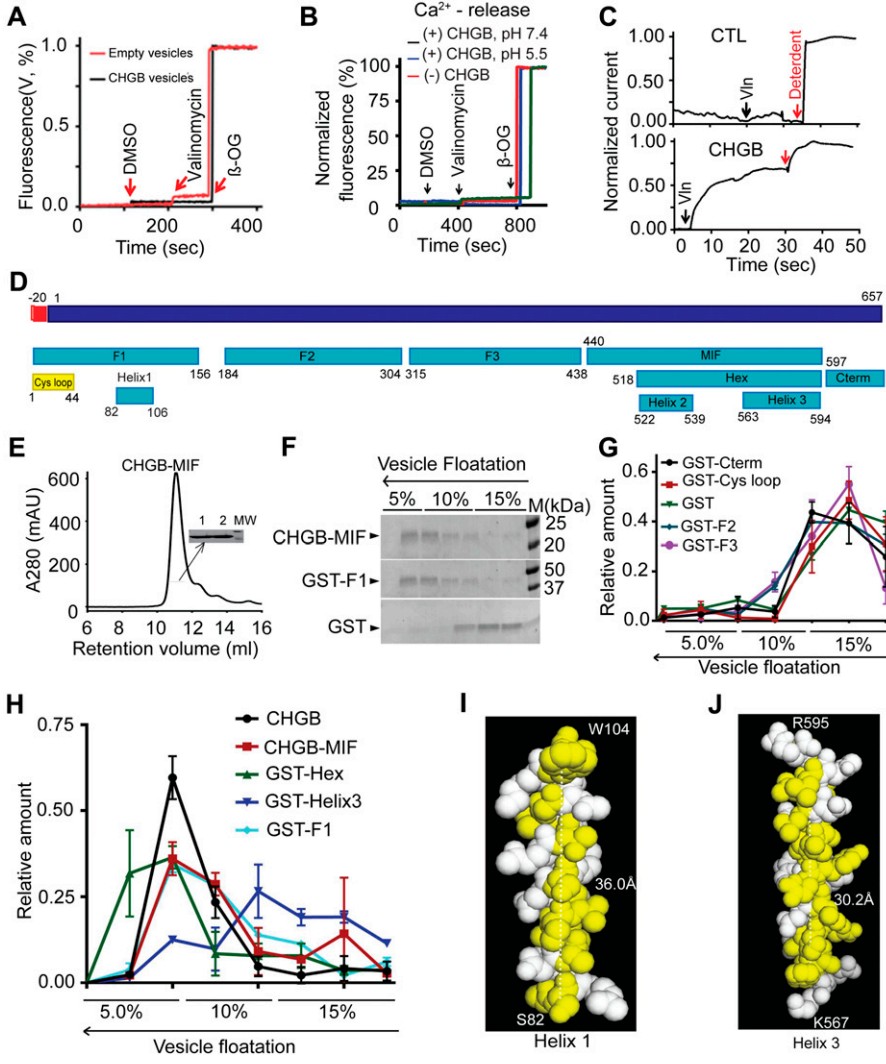

**Figure 3. Mapping amphipathic helical segments for CHGB insertion in membrane.**
**(A)** CHGB vesicles have no membrane cracks. CHGB vesicles loaded with 1.0 µM carboxy fluorescein were treated with 1.0 µl DMSO, 1.0 µM valinomycin, and finally with 5.0 mM β-OG to disrupt vesicles. **(B)** CHGB vesicles leak no $Ca^{2+}$ at pH 7.4 or 5.5. Vesicles loaded with 1.0 mM $CaCl_2$ were treated with 1.0 µl DMSO, 1.0 µM valinomycin, and finally with 5.0 mM β-OG. **(C)** $Cl^-$ release from vesicles recorded with a Ag/AgCl electrode. 300 mM KCl inside vesicles; 300 mM K-isethionate and 0.2 mM KCl, 10 mM Hepes pH 7.4 outside. 0.25 µM valinomycin started the release. At the end 10 mM β-OG was added to release all $Cl^-$. A typical trace out of three (bottom) was shown. Control vesicles (top panel) did not show valinomycin-triggered response. The recordings were normalized to the β-OG signal. **(D)** A diagram of CHGB and different fragments. **(E)** SEC of CHGB-MIF in a Superdex 200 column. The inset is SDS–PAGE of two peak fractions. **(F)** Vesicle floatation assay for CHGB-MIF, GST-F1, and GST. **(G, H)** Distribution of CHGB and its fragments from vesicle floatation assays as in (F) and Fig S3E. Those staying at the bottom in (G) and those floating in (H). **(I)** A helical model for helix 1 with its hydrophobic residues in yellow. It was energetically optimized in Coot and presented in PyMol. **(J)** Structural model for helix 3.

internal $Cl^-$. The need of valinomycin in these experiments suggested that the CHGB does not conduct $K^+$ significantly.

Chloride release from CHGB vesicles suggests that CHGB in membrane forms an anion conductance, either a transporter or an anion channel. It therefore must have a membrane-spanning domain. MIF revealed by the trypsinization experiment probably contributes to the transmembrane domain. Mass spectrometry and N-terminal sequencing of MIF (Figs 2E and 3D) identified it as CHGB 440-597, which is named as CHGB-MIF. Secondary structure analysis of CHGB-MIF revealed a shorter segment (Hex, CHGB 518-597) containing two α-helices interspaced by a short random-coil loop (Fig 3D and online servers detailed in Fig S4C). Vesicle flotation assays showed that recombinant CHGB-MIF alone interacts with membranes (Fig 3E and F), confirming that it is indeed a major contributor to CHGB-membrane interaction (Figs 1E and 2F).

Systematic mapping of MIFs identified two amphipathic α-helices in CHGB (helices 1 and 3 in Figs 3D and S4A–G). Based on secondary structure prediction and the CHGB post-processing peptides, we purified multiple short segments of CHGB, alone or as GST-fusion proteins, and reconstituted them for vesicle floatation (Figs 3D, F–H, and S3E). These short segments were all prepared as stable and monodisperse proteins without aggregation. As a positive control, $His_6$-tagged MIF floated well (Fig 3E and F) because it was well protected by membrane (Fig 2E). Soluble GST alone failed to float (Fig 3F). Except helix 3, all fragments (Fig 3G and H) fell into two groups: those floating and those sinking (Fig 3G versus H; Fig S3E). GST-Helix 3, containing an amphipathic helix, was distributed between the two (Fig 3H). The N-terminal fragment F1 (Fig 3D and H) contains the Cys-loop, which is a soluble sorting signal for granule biogenesis (Cys-loop Fig 3G), and a long amphipathic helix (helix 1, Figs 3I and S4A, B, and E). A structural model of helix 1 in Fig 3I shows its extensive hydrophobic surface (yellow; ~36 Å long). Not protected from trypsinization of CHGB in egg PC vesicles, helix 1 probably lies on the water membrane surface, instead of penetrating deeply into the membrane. Helices 2 and 3 (Figs 3J and S4C, D, F, and G) are well conserved within the CHGB subfamily (Fig S4C and D) and have weak and strong hydrophobic moments, respectively (Fig S4F and G). Helix 2 is thus hydrophilic

and helix 3 amphipathic. Structural modeling of helix 3 (Fig 3J) revealed a hydrophobic surface (yellow) that is >30 Å in length, sufficient to span a typical bilayer membrane. Possible cooperativity between helices 2 and 3 within the Hex segment (Fig 3D) or between the Hex fragment and helix 1 may contribute to the "tightly membrane-associated form" of CHGB (Pimplikar & Huttner, 1992).

## CHGB alone suffices to form an anion-selective channel

To distinguish whether CHGB forms a transporter or a channel (Fig 3C), we first tested if it is feasible to record its activity in planar lipid bilayers (Lee et al, 2013). The membrane-remodeling property of CHGB made it difficult to record from many CHGB molecules. Instead, when diluted CHGB vesicles of low PLR (≤1:5,000) were fused into planar lipid bilayers in the presence of 0.5 mM CaCl$_2$, we observed multiple channel events (Fig 4A). Under an asymmetrical Cl$^-$

condition (42/166 mM), a significant inward current was recorded at 0 mV. The current was ~0 pA at 30 mV (red trace in Fig 4A), suggesting a reversal potential of ~+30 mV. The ionic conditions (*cis*/*trans*) defined the Nernst potentials for Na$^+$ (173/167 mM), Cl$^-$ (42/166 mM; including a small amount of F$^-$), MES$^-$ (10/10 mM), and Ca$^{2+}$ (0.5/0.001 mM; considering a trace amount of ~1.0 μM Ca$^{2+}$ in MilliQ water) to be 0.4, 34.4, 0, and 315 mV, respectively. The average current versus potential plot (Fig S5C) suggested a reversal potential of +30.5 mV. Linear fitting of the relation between single-channel currents and transmembrane potential revealed a single-channel conductance of ~140 pS and a reversal potential of ~29.9 ± 2.7 mV (red line in Fig 4B). The measured reversal potentials are close to the Nernst potential of Cl$^-$. Under the assumption of zero conduction for MES$^-$ and Ca$^{2+}$, the estimated permeation ratio between Na$^+$ and Cl$^-$ ($P_{Na}/P_{Cl}$) is ~1 : 12.6, suggesting that the channel favors Cl$^-$ over Na$^+$ significantly. The trace conduction of MES$^-$ and Ca$^{2+}$ might

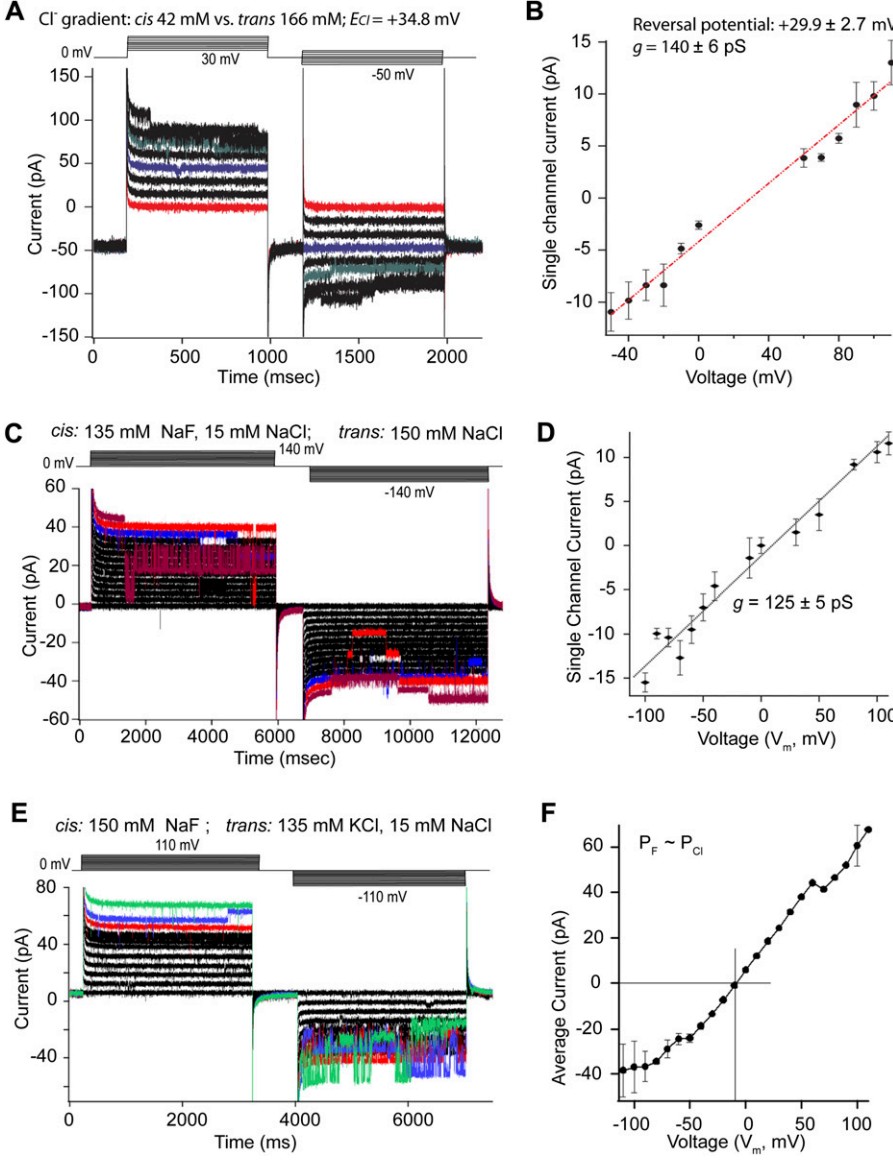

**Figure 4. CHGB forms a chloride channel in membrane.**
**(A)** Electrical recordings from multiple channels in a planar lipid bilayer. Solutions were *cis*: 30 mM NaCl, 130 mM Na-isethionate; 1.5 mM NaF, 0.5 mM CaCl$_2$, 10 mM MES-Na/HCl (9.4 mM Cl$^-$), pH 5.5; and *trans*: 157 mM NaCl, 10 mM MES-Na/HCl (9.4 mM Cl$^-$), pH 5.5. Holding potential: 0 mV; pulses in 10 mV steps from 30 to 110 mV and from 30 to −50 mV in two epochs, respectively. The Nernst potential for Cl$^-$ (41.9 mM/166.4 mM; $E_{Cl}$) is 34.4 mV. At 0 mV, the current is ~ −50 pA. **(B)** Single-channel currents versus transmembrane voltage ($V_m$) from three different membranes in the same ionic conditions as in panel A. Error bars: SD from 3 to 10 histogram-based measurements of single-channel currents under each $Vm$. Fitting of the data with a linear function (dotted line) yielded a single-channel conductance of ~140 ± 6 pS and a reversal potential of +29.9 ± 2.7 mV. **(C)** Electrical recordings from multiple channels in a planar lipid bilayer. Solutions were *cis*: 135 mM NaF, 15 mM NaCl, 0.5 mM CalCl$_2$, 5.0 mM MES-Na/HCl (~4.7 mM Cl$^-$) pH 5.5; and *trans*: 150 mM NaCl, 5.0 mM MES-Na/HCl pH 5.5. Holding potential: 0 mV; pulses in 10 mV steps to 140 and −140 mV in two epochs, respectively. Recordings at ±140, ±130, and ±120 mV were colored dark brown, red, and blue, respectively. **(D)** Single-channel currents versus transmembrane voltage ($V_m$). Error bars: SD from three to five histogram-based measurements of single-channel currents under each $Vm$. Fitting of the data with a linear function (dotted line) yielded a single-channel conductance of 125 ± 6 pS with 150 mM F$^-$/Cl$^-$. **(E)** Small macroscopic currents recorded from a bilayer with multiple channels. *cis*: 150 mM NaF, 0.50 mM CaCl$_2$, and 5.0 mM MES-Na/HCl pH 5.5, and t*rans*: 135 mM KCl, 15 mM NaCl, and 5.0 mM MES-Na/HCl, pH 5.5. Holding potential at 0 mV, and pulses in +10 and −10 mV steps (top). Recordings at ±110, ±100, and ±90 mV were colored in green, blue, and red, respectively, to show the channel switching events. **(F)**. Steady state currents from the recordings in (E) plotted against $V_m$. Currents become smaller at negative range, because of more frequent channel closing (error bars are SD, n = 4). The reversal potential ($E_{rev}$ = −8.0 mV) was used to estimate the permeation ratio ($P_F/P_{Cl}$) of F$^-$: Cl$^-$ using a simplified GHK equation, $E_{rev} = -RT/F \ln(P_F [F^-]_o/P_{Cl} [Cl^-]_i)$. The *cis* side is equivalent to the luminal side (outside) where the CHGB resides. From eight measurements, $P_F/P_{Cl}$ = 1.2 ± 0.2.

have made the estimated $P_{Na}/P_{Cl}$ slightly higher than the true value. Empty vesicles and vesicles prepared with CHGA were used as negative controls. Hundreds of bilayer membranes were analyzed to ensure high reproducibility and sufficient statistical power.

Comparing the measured single-channel conductance in Fig 4B with a chord conductance of ~1.5 nS from the small macroscopic currents in Fig S5C suggested that there were ~11 channels in the patch recorded in Fig 4A. The average open probability ($<P_0>$) of the channel remained at ~100% within a range of 60 mV above or below the reversal potential, and dropped to ~70% at 80 mV away from the reversal potential (the red line in Fig S5C). These patches lasted for 10–90 min, suggesting stable CHGB channel function. Under the ionic settings in Fig 4A, we found that NaF and 0.5 mM $CaCl_2$ in the *cis* side helped to stabilize the bilayer membranes after vesicle fusion and minimize leak currents. When lipid mixtures known to have phase separation were used for CHGB reconstitution, for example, 1-palmitoyl-2-oleoyl-sn-glycero-3-phosphocholine (POPC) : brain sphingomyelin : cholesterol = 3:1:1 in weight ratio, we sometimes observed a nonspecific leak channel with a single-channel conductance of ~225 (±5) pS. Because the average single-channel conductance of CHGB is significantly smaller than 225 pS, it was quite easy to remove contaminating events from these leak channels by single-channel analysis. As negative controls, vesicles prepared with BSA, CHGA (Fig S4I), a CHGB deletion mutant lacking the CHGB-MIF (CHGBΔMIF), and CHGB-MIF itself all failed to generate any channel activity. A strong selectivity of anions over cations and the average $P_0$ over $Vm$ ruled out the possibility that the measured conductance was contributed by non selective lipid ion channels previously observed in bilayers made of lipid mixtures that tend to exhibit phase separation (Mosgaard & Heimburg, 2013). We therefore conclude that the observed anion channels were made by CHGB in membrane.

When we recorded the CHGB channels under symmetrical anions of ~159 mM $F^-$ or $Cl^-$, the measured single-channel conductance of CHGB is ~125 pS (Fig 4C and D), fairly close to the 140 pS measured under asymmetrical anion conditions (Fig 4B). The small difference in single-channel conductance probably came from the different ionic conditions (Fig 4C versus 4A). When analyzing the recordings in Fig 4A and C, we determined the dominance of the CHGB anion channels based on two criteria: (i) the membrane patches should show a positive current ($Cl^-$ influx) at 0 mV when 15 mM NaCl (or KCl) was present on the *trans* side for inducing vesicle fusion; and (ii) after salt balancing, the bilayer recordings at high voltages should show events that briefly reached 0 pA when all channels were closed (as in Fig S5A). Besides, our recordings showed subconductance states (as highlighted in Fig S5B), which will need detailed characterization in the future.

Comparison of recordings in Fig 4A and B and those in Fig 4C and D suggests that the anion conduction of the CHGB channel could probably be sensitive to the anion concentrations on both sides of the membranes. To further evaluate the $Cl^-$ selectivity and anion sensitivity of the CHGB channel, we recorded single-channel currents with asymmetrical [$Cl^-$] and symmetrical [$K^+$] at pH 5.5 (*cis/trans*: 1.5 / 15 mM $Cl^-$ versus 151.5 / 150 mM $K^+$; Fig S5D and E, where an MES buffer was used without adding more $Cl^-$ during pH adjustment). K-isethionate was added to maintain high osmolality and ionic strength. Under such conditions, the recorded channels

had no measurable outward current in the positive voltage range, suggesting no detectable conduction of $K^+$. Analysis of the average single-channel currents at different negative voltages yielded an estimated chord conductance of ~58 pS (black line in Fig S5D, for 15 mM $Cl^-$) and, by extrapolation, a reversal potential of ~+65 mV, close to the calculated Nernst potential ($E_{Cl}$ = +58.4 mV) of $Cl^-$, but far away from that (0 mV) of $K^+$, $MES^-$, $H^+$, or $OH^-$. The estimated reversal potential, being more positive than $E_{Cl}$, suggested that the CHGB channel had a very low permeation of $K^+$ and the $P_K/P_{Cl}$ might be much lower than the $P_{Na}/P_{Cl}$ measured from data in Fig 4B. The CHGB channel is thus $Cl^-$-selective with very low $K^+$ conduction and its anion conduction is sensitive to [$Cl^-$] as expected for a $Cl^-$ channel.

When we measured reversal potentials of the CHGB channels in bi-ionic conditions, the estimated permeation ratio between $F^-$ and $Cl^-$ (Fig 4E and F) is ~1.2. Good permeation of $F^-$ suggested possible conduction of $OH^-$, which, however, should be very low in comparison with $Cl^-$ flux because the CHGB channel is sensitive to the concentration of the permeating ions in the range of 1.5 to 150 mM. The $OH^-$ concentration of $1.0 \times 10^{-8.5}$ or $1.0 \times 10^{-7.0}$ M at pH 5.5 or pH 7.0, which is six to seven orders of magnitude lower than [$Cl^-$], would make its permeation across the CHGB channel negligible even though change of pH might modulate the channel activity. For reliable electric recordings in bi-ionic settings, we also checked that in higher voltage ranges (±80 to ±130 mV) closure of all channels reached zero current level briefly (as in Fig S5A) such that the reversal potentials determined from the current–voltage curves (Fig 4F) were reliable for estimating the bi-ionic permeation ratio.

Because only a small number of channels were measured in lipid bilayers, we questioned if a trace amount of contaminating anion channels might have contributed to the recorded activities. We separated 20 μg of purified CHGB that had been kept in a cold room for ~4 d, and detected two faint smaller bands (1 and 2 in Fig S1B). When the full-length CHGB band and the two shorter bands were cut out and digested for HPLC/MS analysis and proteomic identification (Tables S1 and S2), all three bands were heavily dominated by CHGB peptides. The two smaller bands shared five peptides and were therefore CHGB degradation products. Other candidates had much fewer matched peptides, less than six for the CHGB band and only one for two smaller bands. Among them, only KvQT member 5 (accession number E9Q9F in Table S1) is a $K^+$ channel, but without an anion channel. When we compared the scanned density of the bands (right panel in Fig S1B), the two degradation bands, usually absent in fresh samples (Fig S1A), were ~1.5% of total CHGB. Results from diluted proteins in the same gel suggested that any other contaminant that is more than 0.12% of the total mass would have been detected. The purified CHGB thus had >99.8% purity, making it very unlikely (<0.12%) for a contaminating anion channel to account for the observed channel activities in lipid bilayers.

## CHGB conducts $F^-$ and $Cl^-$ better than other anions

Because it was critically important to stringently rule out the possibility of trace contaminant channels yielding the measured channel activity in Fig 4, we quantified anion flux from a large number (>$1.0 \times 10^{11}$) of CHGB vesicles. The Ag/AgCl measurement in

Fig 3C was limited to Cl⁻, not other anions; nor was it highly stable because of slow mixing of valinomycin and drifts of electrode potential. We instead implemented a light scattering–based flux assay (Fig S3F) (Stockbridge et al, 2013) by both steady-state and stopped-flow fluorimetry. In both systems, valinomycin triggers K⁺ efflux and anion release via CHGB from vesicles, resulting in a sudden drop of intravesicular osmolality and collapse of vesicles into ellipsoids that scatter more light (Fig S3F). Extrusion was performed to control vesicle dimensions so that large-sized vesicles would not dominate the measured signals. As a positive control, the bacterial Cl⁻/H⁺ cotransporter (EriC; 2:1 exchanger) in vesicles yielded a robust increase in light scattering (Fig S4H) (Stockbridge et al, 2013) when compared with empty vesicles. The CHGB vesicles prepared in parallel delivered a strong signal (Figs S4H and 5A) at both pH 7.4 and 5.5. The increase in steady-state light scattering saturated within the 10-s break for adding valinomycin, slightly quicker than the 20-s duration of Cl⁻ efflux in Fig 3C. A nonspecific Cl⁻-channel blocker, DIDS (5,4′-diisothiocyanatos-tilbene-2,2′-disulfonic acid), was able to block the strong flux signal with an apparent $k_D$ = 0.9 μM (Fig 5B), reflecting an affinity which is one to two orders of magnitude higher than the DIDS affinity for all ClC-family proteins.

As negative controls, recombinant CHGA in vesicles produced no signal (Fig S4I and J); nor did CHGB-MIF or CHGBΔMIF (Fig 5C and D).

Hence, CHGB-MIF alone is insufficient to form the anion channel although it is indispensable to a functional channel. Considering the negatively charged residues contributing to anion selectivity of other Cl⁻ channels (Maduke et al, 2000), we deleted a short loop (residues: 540–551; CHGB-ΔL1) right after helix 2 (Fig S4C) and mutated three negatively charged residues in the loop to Ala (marked with arrows in Fig S4C and D). The same amount of different mutant proteins was reconstituted into the same amount of (5 mg/ml) lipids to allow comparison among them by the flux assay (Fig 5E). CHGB-ΔL1 had significantly lower flux than wild-type CHGB. Mutation of the well-conserved E558 blocked approximately 40% of the flux signal, whereas neither E545A nor E552A incurred significant effect (Fig 5E). These data further demonstrate that helix 3 and the loop ahead of it are important for ion conduction of the CHGB channel (Fig S4D).

We next compared the relative ion selectivity of the CHGB channel by loading different anions into vesicles for flux assays. Our data show that the CHGB channel conducts Cl⁻ and F⁻ much better than Br⁻, I⁻, NO₃⁻, SCN⁻, formate, or citrate (Fig 5F), suggesting that under the conditions, especially the high positive potential inside the vesicles, for the light scattering–based flux assays, the CHGB channel is more selective than the other well-known Cl⁻ channels or transporters (Maduke et al, 2000; Fahlke, 2001). The equally robust light scattering–based flux signals for F⁻ and Cl⁻ are consistent with the measured permeation ratio $P_F/P_{Cl}$ = 1.2 (Fig 4F).

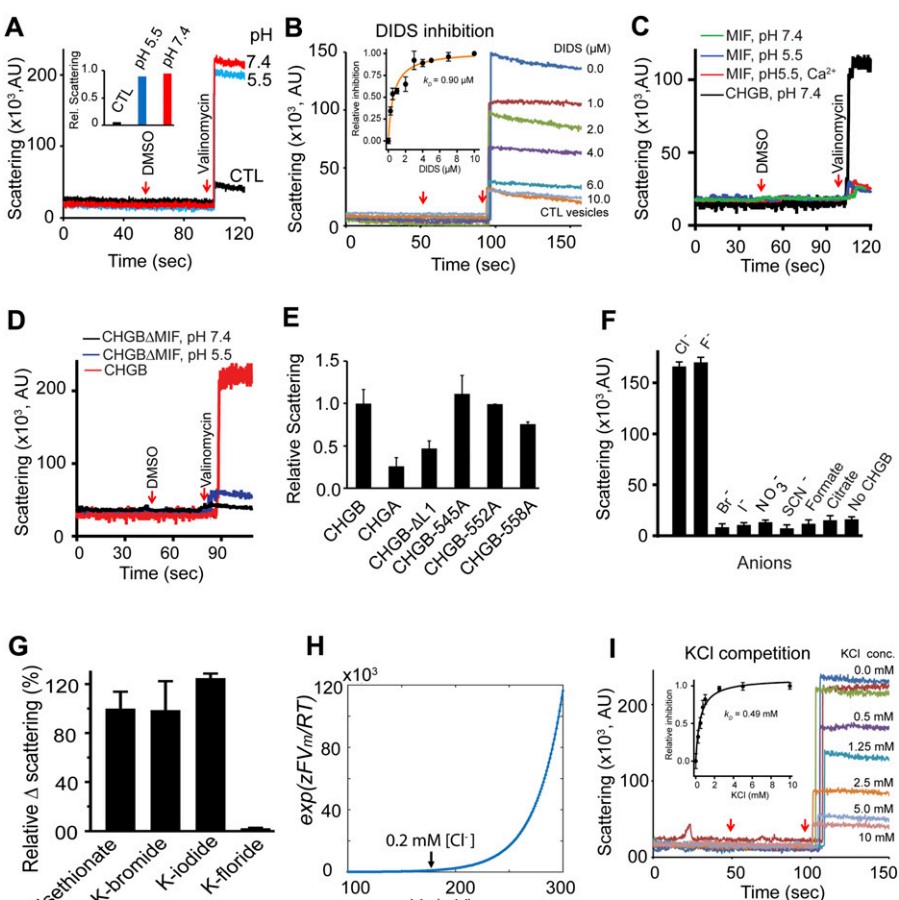

**Figure 5. CHGB conducts F− and Cl− better than other anions.**

**(A).** Light scattering–signals measured from vesicles with (red and blue for pH 7.4 and 5.5, respectively) and without (black) CHGB. A 30-s pause for adding DMSO or valinomycin was not included. **(B)** The CHGB proteoliposomes were treated with different concentrations of the DIDS for 5 min at RT before experiment. DIDS inhibits CHGB-mediated Cl⁻ flux. Red arrows indicate the addition of DMSO and valinomycin. Inset: change in the scattering was plotted against the DIDS concentration. Fitting with a Hill equation yielded a $k_D$ = 0.9 μM; n = 1.0. **(C)** Flux assay from vesicles with and without CHGB-MIF under different conditions in comparison with CHGB vesicles (black). **(D)** Flux assays of liposomes with and without CHGBΔMIF at two different pH's. Wild-type control in red. **(E)** Flux assays of liposomes containing CHGA, CHGB, and CHGB mutant proteins. Residues E545, E552, or E558 in the loop (L1) between helix 2 and helix 3 were mutated to alanine. **(F)** Relative anion selectivity of CHGB channel based on the flux assay. Error bars represent SD from three different sets of experiments. **(G).** Instead of 300 mM K-isethionate, 300 mM KF, KBr or KI was introduced to the extravesicular side against 300 mM KCl-loaded vesicles. Br⁻ or I⁻ supported significant flux signal. But, 300 mM F⁻ abolished the flux. Error bars: SD, n = 3. **(H).** Calculated correction factor for valinomycin-mediated K⁺ transport because of steady state Nernst potential for 300 mM KCl–loaded CHGB vesicles with 0–5 mM KCl outside. **(I)** Flux assay from 300 mM KCl–loaded CHGB vesicles in the presence of 0–10 mM KCl and 300 mM K-isethionate in the extravesicular side. Vesicles had 1:10,000 PLR of CHGB: egg PC. Three independent sets of vesicles prepared from separate cell cultures (n = 3) were used to ensure high statistical power. Inset: change in the scattering was plotted against the KCl concentration and fitting with a Hill equation yielded a $k_D$ = 0.49 mM and n = 1.0.

The ion selectivity data (Fig 5F) make two predictions. One is that extravesicular 300 mM $Br^-$ or $I^-$ should not affect the light scattering–signals, whereas $F^-$ should completely stop it, precisely what our experiments showed (Fig 5G). The other is that a low concentration of extravesicular $Cl^-$ should shift the initial steady-state Nernst potential of $Cl^-$ ($E_{Cl}$) across vesicle membranes and significantly inhibit $Cl^-$ flux because of a voltage-dependent factor affecting the rate of valinomycin-mediated potassium flux (Fig 5H; $exp\ [z\delta FV/RT]$; more details in Ion flow through the CHGB requires a fast-conducting channel, not a slow-acting transporter section). Indeed, extravesicular $[Cl^-]$ of 0.1 to 2 mM significantly impaired the light scattering–signal with an apparent $k_D$ = 0.49 mM (Fig 5I).

### High cooperativity among CHGB subunits in forming functional channels

Steady-state fluorimetry measurement cannot detect fast channel-opening events because of slow and uneven equilibration of valinomycin among vesicles. A stopped-flow fluorometer (Hayner et al, 2014) with a dead time of 2 ms was used to overcome this limit through fast mixing. We compared the light scattering–signals by both steady-state and stopped-flow fluorimetry, and titrated PLRs of CHGB vesicles with a fixed lipid concentration of ~0.4 mg/ml. When PLR < 1:50,000, no signal was detected in either system (Fig 6A and B). When PLR reached 1:1,000, the signal approached its maximum. All stopped-flow traces showed instantaneous jumps, suggesting a fast efflux of $K^+/Cl^-$ within 2 ms. Because of fast and even partitioning of valinomycin into individual vesicles, the instantaneous jumps in light scattering reflected a sudden change in vesicle shape after $K^+/Cl^-$ efflux. When the measurements during the first 5 s from the steady-state fluorimetry and the first 40 ms of the stopped-flow data (Fig 6C and D) were plotted against [CHGB] and fitted with a Hill equation (Fig 6E), the estimated Hill co-efficients were ~1.2 from the steady-state measurement and ~4.2 from the stopped-flow data, suggesting high positive cooperativity among CHGB subunits in forming a functional channel. The steady-state data revealed less cooperativity probably because of uneven and slow mixing of valinomycin with vesicles and the complications resulted from water movement in response to osmolality change. When the data in Fig 6E were plotted against the average number of CHGB subunits per 100 nm vesicle (Fig 6F), the threshold for strong signals occurred at ~4 CHGB subunits per vesicle, and nearly 80% of the maximum signal was achieved with ~8 CHGB subunits per vesicle.

Random reconstitution and a Poisson distribution suggest that the CHGB channel is a tetramer. CHGB dimers were randomly distributed among individual vesicles. According to the parameters for our vesicles (Table S3), a PLR = 1:10,000 yielded an average number ($\lambda$) of CHGB monomers per 100-nm vesicle of ~8. The Poisson distribution allows us to evaluate the fraction of vesicles that might not have enough CHGB subunits to form a functional channel. Given that $P(k)$ represents the probability of vesicles containing $k$ copies of CHGB monomers, where $k$ is a discrete number (0, 1, 2, 3, ...) and $\lambda$ the average number of CHGB monomers per vesicle that was set by adjusting the PLR of the CHGB vesicles, we calculated the fractions of vesicles, $P(k < N)$, that would each have less than $N$ copies of CHGB monomers with $N$ = 2, 3,..., 8. If a functional channel needs N subunits, then $P(k < N)$ represents the theoretically predicted fraction of

vesicles having no functional channel. The predictions were listed together with experimental data (last column) of Table 1. Least squares analysis of the experimental data and $P(k < N)$ (total variances in Table 1) found N = 4 (bold in Table 1), meaning that each channel should have four subunits (more details in A Poisson distribution of CHGB subunits among vesicles predicts tetrameric stoichiometry of functional channels section). Given the stable CHGB dimers in detergents, a channel is more likely to have an even number of subunits. A tetramer is thus probably a dimer of two dimers.

The strong signals in the light scattering–based flux assays predict that valinomycin may form a $K^+$-conducting channel to match the fast anion flow. The quick, sudden change in vesicle shape requires fast $K^+$ flux within the 2 ms mixing time. Varying [valinomycin] per vesicle was hence expected to affect the signal. Titration of [valinomycin] in the stopped-flow measurement (Fig 6G) found that when [valinomycin] <0.25 $\mu$M, the flux signal was almost non detectable, whereas 2–5 $\mu$M delivered the maximal signal. Fitting of the dose response data with a Hill equation (Fig 6H) identified a Hill coefficient of $n$ = 2.6, indicating high positive cooperativity among valinomycin molecules in moving $K^+$ across membrane. These data accord well with our observation that 0.1 $\mu$M valinomycin triggered no detectable flux signal. The cooperativity suggests that valinomycin function as trimers or high-order oligomers in membrane, reminiscent of the dimeric valinomycin channel in ultrathin membranes (Gliozzi et al, 1996). To match the $Cl^-$ flux, each valinomycin oligomer must have a conductance >3 pS, a lower limit for an uncharacterized valinomycin channel.

# Discussions

### CHGB-membrane interaction and formation of anion channel

Our data collectively support a working model that CHGB inserts itself into lipid membranes, which promotes CHGB oligomerization by shifting a dynamic equilibrium between the dimers and the predicted tetramers, and generates a functional $Cl^-$ channel by spanning two leaflets of a biomembrane (Fig 7). Even though we cannot exclude the possibility that the CHGB organizes the lipids to form a lipid ion channel, the nonselective nature of previously reported lipid ion channels and the anion selectivity of the CHGB channel (Figs 3A–C, 4A, B, S5C, and 5F) together suggest that the CHGB channel is probably formed by the protein in the presence of the lipids. This explanation also agrees with the observations that the CHGB needs no specific lipids to form the anion channel because the flux assays found that the channel functions well in 1,2-dio-leoyl-sn-glycero-3-phosphocholine (DOPC), 1-palmitoyl-2-oleoyl-sn-glycero-3-phosphocholine (POPC), egg phosphocholine (PC), 1-palmitoyl-2-oleoyl-sn-glycero-3-phosphoethanolamine (POPE)/1-palmitoyl-2-oleoyl-sn-glycero-3-phosphoglycerol (POPG) (POPE/POPG = 3:1 weight ratio) mixture, or DOPC/sphingomyelin/cholesterol mixture. Because eukaryotic cells have sufficient ER membranes, nascent CHGB molecules should be preferentially membrane-associated from the luminal side. Both helices 1 and 3 are involved in the membrane insertion (Fig 3I and J) and the loop before helix 3 is important for anion conduction (Fig 5E). Moreover, the CHGB-membrane interaction may

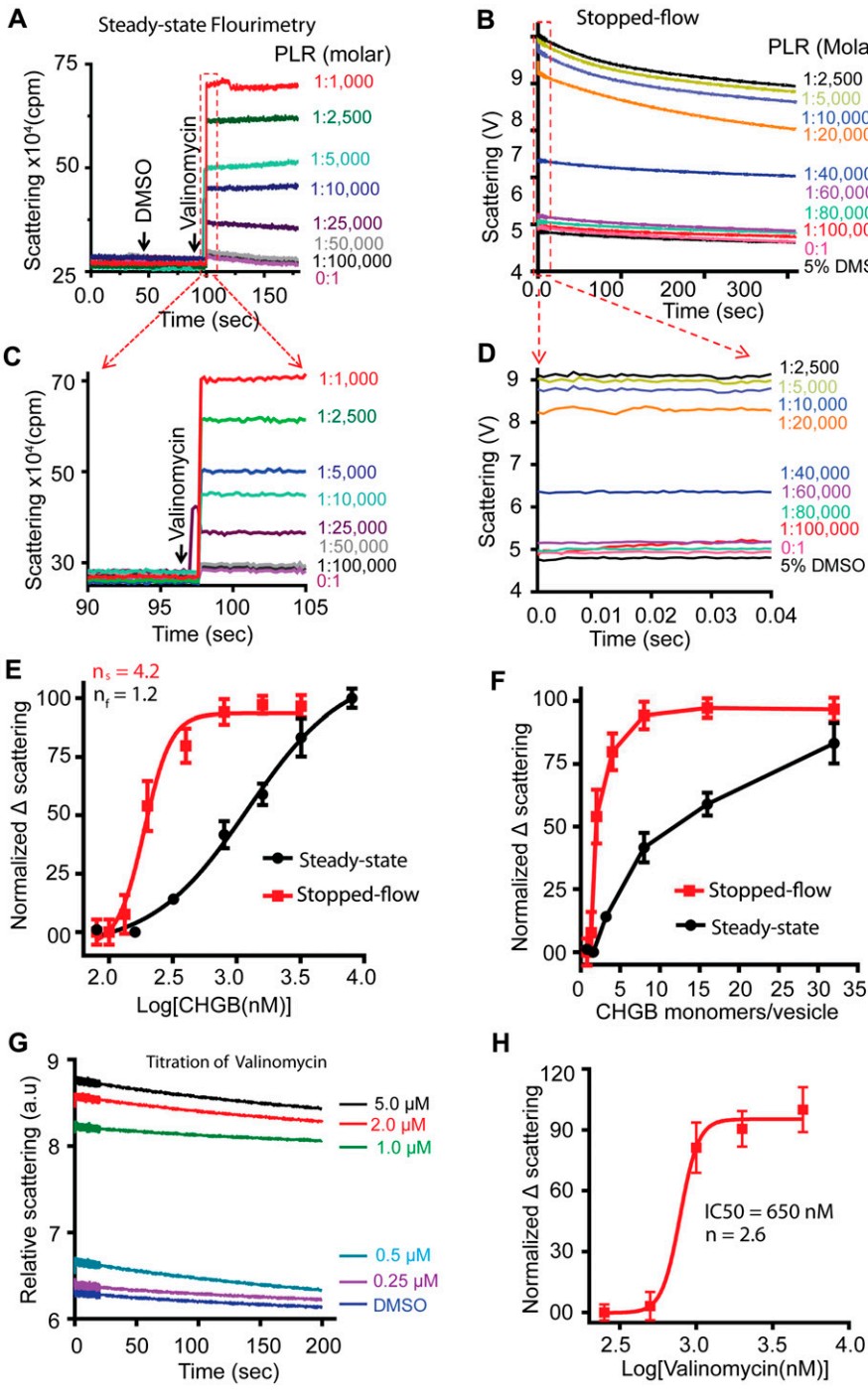

**Figure 6. Fast kinetics of Cl⁻ release from the CHGB channels in vesicles.**

**(A)** 300 mM KCl–loaded vesicles changed into 300 mM K-isethionate right before the assay. Time-lapsed light scattering was measured at 600 nm in a Fluoromax-4. Vesicles of different PLRs were prepared in the same batch. Adding DMSO triggered no signal, but adding 1.0 μM valinomycin did. **(B)** Time-lapsed light scattering (610 nm) measured in a stopped-flow system, by mixing equal volumes of vesicles and 2.0 μM valinomycin in buffer. 5% DMSO solution was added as negative control. Vesicles of PLR = 1:1,000 used to set a proper accelerating voltage for the PMT tube to have the broadest dynamic range. **(C)** Expanded region in the red box in panel A. **(D)** Expanded region in the red box in panel B. Data from the first 40 ms after the dead time of ~2 ms for mixing are shown. **(E)** Average signals from the first 5-s data in (C) (black) and those from the first 40 ms in (D) (red) are plotted against [CHGB]. Fitting with Hill equations (solid lines) generated different Hill coefficients. Steady state data from four different datasets (n = 4; error bars are SD); Stopped-flow data from two sets of triplicate measurements (n = 6; error bars are SEM). **(F)** The data in panel E were plotted against the average number of CHGB monomers per 100-nm vesicle. The threshold for generating a significant signal is ~4 monomers per vesicle for both measurements. **(G)** Titrating valinomycin for the light-scattering from CHGB vesicles in the stopped-flow system. The CHGB vesicles had PLR = 1:25,000. More vesicles than those in panel B were used to boost the signal, which also increased the baseline signal (buffer + DMSO). Three triplicate measurements were obtained (n = 9; error bars are SEM). **(H)** The normalized signal from panel G were plotted again [valinomycin] and fitted with a Hill equation to yield a Hill coefficient of ~2.6.

promote high-order oligomeric forms in a calcium-dependent way (Fig S2B–F) and can remodel the bilayer membranes into nanostructures (Fig 2F). For simplicity, we depict a channel pore with its two opposing monomers, omitting other subunits in Fig 7. Once a channel is formed, it opens most of time (Figs 4 and S5C), making its resident membrane highly permeable to Cl⁻, but not other anions except F⁻. Inside the secretory granules, such high selectivity would be useful in keeping metabolic anions from becoming concentrated into the secretory granules and being dumped as waste.

## Membrane-insertion induces CHGB channel formation

Our data show that with sufficient lipids both CHGB and CHGA are fully reconstituted in the liposomes (Figs 1E and S4I), but CHGA does not form a channel. We hence propose that the full-length CHGB may be preferentially in membrane in the regulated secretory pathway. Because both ER and Golgi membranes have their own Cl⁻ channels, the CHGB channel is probably nonessential for ER and Golgi. The Cys-loop domain of the CHGB guides its delivery to the secretory granule

**Table 1. A Poisson distribution of CHGB subunits among vesicles and the most probable stoichiometry of a functional channel.**

| | Fraction of vesicles have fewer than N monomers, $P(k < N)$ | | | | | | | Stopped-flow data |
|---|---|---|---|---|---|---|---|---|
| | 2 | 3 | 4 | 5 | 6 | 7 | 8 | |
| $\lambda$, <CHGB>/vesicle | | | | | | | | |
| 0.8 | 0.81 | 0.95 | 0.99 | 1 | 1 | 1 | 1 | 1 |
| 1.67 | 0.503 | 0.765 | 0.911 | 0.972 | 0.993 | 0.998 | 1 | 0.912 |
| 2 | 0.406 | 0.677 | 0.857 | 0.947 | 0.958 | 0.983 | 0.996 | 0.846 |
| 4 | 0.092 | 0.238 | 0.434 | 0.629 | 0.785 | 0.889 | 0.949 | 0.481 |
| 8 | 0.003 | 0.014 | 0.042 | 0.1 | 0.191 | 0.313 | 0.453 | 0.19 |
| Total variances | 0.583 | 0.143 | **0.024** | 0.044 | 0.112 | 0.208 | 0.318 | |

Bold value represents the optimal fitting for the experimental data.

where it executes its specific intracellular functions as investigated in a separate paper (Yadav et al, unpublished), which studies the function of "tightly membrane-associated" CHGB on the cell surface (Pimplikar & Huttner, 1992) and connects the CHGB channel function with granule maturation ex vivo and in vivo.

Membrane insertion of an amphipathic protein without canonical transmembrane segments to form ion channels has multiple precedents: Hemolysin, C-type lectin, VopQ, etc., are a few (Ramarao & Sanchis, 2013; Sreelatha et al, 2013; Mukherjee et al, 2014). Thermodynamically, oligomerization of individual subunits and interactions among them and with lipids increase entropy of water molecules liberated from hydrophobic surfaces, such as those from helices 1 and 3 in CHGB (Fig 3I and J), and overcome the energy cost of dehydration and membrane insertion.

The observations of both heat-stable, soluble fractions and "tightly membrane-associated form" of CHGB may reflect two different biochemical conditions of CHGB (Benedum et al., 1986, 1987; Pimplikar & Huttner, 1992; Yoo, 1995b). CHGB-coated 25 nm nanoparticles may reconcile these observations. These nanoparticles are densely packed with proteins and are expectably heat-stable and soluble by burying CHGB's hydrophobic domains. It would be interesting to examine whether CHGB in the soluble (or heat-stable) fractions takes similar structures as the 25 nm nanoparticles (Fig 2B–D). Similarly, structures of CHGB dimers, tetramers, and higher order oligomers will be needed to reveal their allosteric changes during channel formation. Further, the sidedness of CHGB insertion in membrane endows special functions such as membrane remodeling and pH- or $Ca^{2+}$-dependent regulation, which await more investigations.

A "tightly membrane-associated form" of the CHGB is resistant to treatment of basic pH, but soluble by Triton X-114 (Pimplikar & Huttner, 1992). Beside a good way to retain the full-length protein and separate it from the CHGB peptides, such a mechanism is consistent with our working model because calcium-induced CHGB aggregates (Fig S2B–F) or CHGB-packed nanoparticles might send some full-length CHGB proteins into the lumen through membrane remodeling. It remains unknown what physiological roles the 25-nm particles, if they form inside cells, would play and how the CHGB in the luminal side is distributed and regulated.

Nevertheless, the CHGB channel activity is an intrinsic property because of the channel activity in different membrane systems tested so far. Considering the highly conserved helices (Fig S4B and D) and the key residues (e.g., E558 in Figs 5E and S4D), we propose that $Cl^-$ conductance represents a universal property within the CHGB subfamily of granin proteins. CHGB channels are quite different in sequence, structure, and functional properties from the six families of chloride channels—CTFR, the large family of glycine/GABA receptor channels, the ClC-family of channels/transporters, the volume-regulated anion channels, the uncharacterized large conductance $Cl^-$ channels, and the $Ca^{2+}$-regulated $Cl^-$ channels of TMEM16A, TMEM16B, and bestrophins (Qu et al, 2006; Duran et al, 2010; Hwang & Kirk, 2013; Pedersen et al, 2016). CHGB channels do not contain the canonical transmembrane $\alpha$-helices; nor the beta-barrel structure of VDAC (voltage-dependent anion conductance) in mitochondria. They have better anion selectivity, higher DIDS-binding affinity, and larger conductance. These make the CHGB channels a new family of $Cl^-$ channels, but a very distant one from the others.

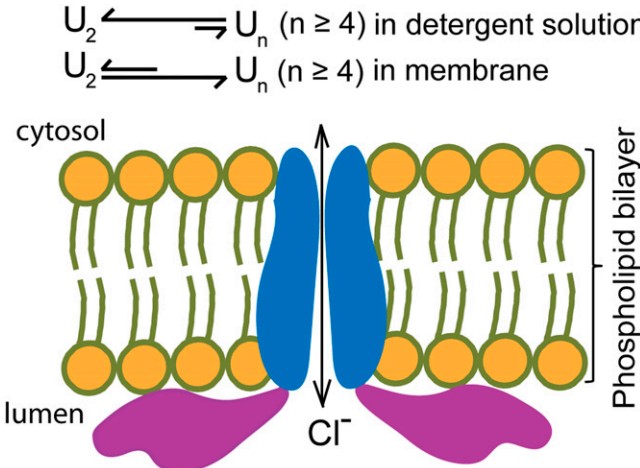

**Figure 7. A working model for CHGB membrane insertion and channel formation.**
Dynamic equilibrium exists between CHGB dimers and oligomers (probably tetramers, $U_n$). In detergents, dimers are favored while in membrane oligomers (tetramers) are. Bottom: a scheme for CHGB insertion into membranes from the luminal side, forming a $Cl^-$ channel and part of each CHGB subunit resides on the luminal surface.

### Two forms of CHGB in native tissues and their relation to CHGB ion channel

CHGB has been prepared in both soluble and membrane-bound fractions in the past (Benedum et al, 1987). The amphipathic nature of CHGB makes it feasible to distribute between these two states, similar to the observed transition of melittin from soluble tetramers to membrane-integrated oligomers that lead to hemolysis or the oligomerization of soluble monomeric human C-type Lectin RegIIIβ and RegIIIγ in membranes that leads to membrane breakdown in $G^-$ and $G^+$ bacteria (Laine et al, 1988; Miki et al, 2012; Mukherjee et al, 2014). The soluble form of CHGB in the secretory granules is likely dominated by CHGB dimers and/or their complexes with themselves, other proteins, or lipids. It may contain partially processed CHGB by proteolysis. The membrane-bound form is very likely dominated by tetramers or high-order oligomers. These two forms together may reconcile the conflicting reports in literature and reveal new physiological properties of CHGB in two different states.

## Conclusions

CHGB is capable of inserting itself into membranes and forming a $Cl^-$ channel with anion selectivity higher than other known families of $Cl^-$ channels. The unique amphipathic features of the CHGB subfamily proteins make them a new family of $Cl^-$ channels that function in the regulated secretory pathway.

## Materials and Methods

### Purification of recombinant CHGB and its mutants expressed in *sf9* cells

We used baculovirus to over-express CHGB proteins (Invitrogen). Preparation of other constructs followed the same general procedure unless separately stated. The cDNA of mouse CHGB gene (mChgb) was cloned from a pcDNA3.1 plasmid (a gift from Dr. Barbara Ehrlich at Yale University) into the pFastBac 1 vector (Invitrogen). Cloning was monitored by mapping with restriction endonucleases and PCR-based sequencing. *Escherichia coli* clones with the recombinant bacmids were isolated after transformation of *E. coli* DH10Bac cells with mChgb/pFastBac1 plasmid and blue/white-screening of the transformed colonies. The bacmid DNA was used to transfect monolayers of *sf9* cells using CellFECTIN II reagent (Invitrogen). Recombinant viruses were harvested 72–96 h after transfection (P1), and further amplified twice to obtain higher titer viruses (P2 and P3). *Sf9* cells in SFM-900 II with 2% vol/vol heat-inactivated FBS and 1× penicillin/streptomycin were infected with recombinant viruses with MOI of one. Cells were harvested 48 h after infection, and lysed in a buffer made of 50 mM Tris, pH 8.0, 10% glycerol, 5.0 mM DTT, 1.0 mM EDTA, 1.0 mM PMSF, 2.0% NP40, 1× protease-inhibitor mix (Sigma-Aldrich), and 1.0 μg/ml each of leupeptin, pepstatin, and aprotinin. After 1 h extraction in the cold room, cell lysates were centrifuged at 100, 000 g for 1 h at 4°C.

The clear supernatant was collected and applied directly to a pre-equilibrated Q-sepharose FF column (Amersham Biosciences). The column was washed with buffer A (20 mM Tris, pH 8.0, 100 mM NaCl, 2.0 mM β-ME, 0.050% Triton X-100, and 0.5 mM PMSF), and eluted with a three-step gradient of buffer B containing 2.0 M NaCl. CHGB-containing fractions were pooled. 25% ammonium sulfate (final) was added to precipitate CHGB protein. The pellet was collected by centrifugation and dissolved in buffer C containing 20 mM Tris, pH 8.0, 100 mM NaCl, 2.0 mM β-ME, 0.050% Triton X-100 of reduced absorbance, and 0.25 mM PMSF. The dissolved mixture was applied to a Ni-IDA column (Amersham Biosciences), washed, and eluted with 300 mM imidazole in the buffer. The eluted protein was concentrated and further purified by size-exclusion FPLC (fast protein liquid chromatography) using a Superose 6 10/30 GL column (Pharmacia Biotech). The CHGB fractions were collected and concentrated to ~5.0 mg/ml using a 30,000 MWCO filter (Millipore).

The CHGB protein concentration was estimated by $OD_{280}$ using a calculated extinction coefficient of 82,405 $M^{-1}cm^{-1}$. Purified CHGB was subjected to 10% SDS–PAGE to confirm its purity. For Western blot, the protein bands in an SDS–PAGE gel were transferred to a PVDF membrane, immunostained with monoclonal antibodies, and visualized by chemiluminescence (SuperSignal West Pico; Thermo Fisher Scientific).

### Expression and purification of individual GHGB fragments from bacteria

Different fragments of mouse CHGB, including CHGB-F1, CHGB-F2, CHGB-F3, CHGB-F4, CHGB-Hex, CHGB-Helix2, CHGB-Helix3, CHGB-Cys, and CHGB-Cterm (Fig 3D), were individually subcloned into the pGEX-kg (Novagen) vector containing an N-terminal GST-tag. The fusion proteins were over-expressed in *E. coli* BL21 (DE3) host cells (Novagen). After cell lysis and centrifugation, the protein was affinity-purified with Glutathione Sepharose 4B (GE Healthcare). The eluted protein was further purified by size-exclusion FPLC in a Superdex 200 column (GE Healthcare) in a buffer containing 20 mM Tris–HCl, pH 8.0, 200 mM NaCl, 1.0 mM EDTA, and 1.0 mM DTT with 0.050% Triton X-100.

### Preparation of proteoliposomes

Reconstitution follows closely a published protocol (Lee et al, 2013). 5 or 10 mg of egg PC lipids (Avanti Polar Lipids) was dried in a glass testtube with argon gas, vacuum-treated for 1.0 h, hydrated with autoclaved MilliQ water, and vortexed before being sonicated in an iced water-bath sonicator to make small unilamellar vesicles. N-decyl β-D maltopyranoside (DM; Affymetrix—Anatrace) was added to 40 mM. After 5–6 h at RT, the detergent/lipid solution became almost completely transparent. Lipid/detergents and proteins in detergents were mixed in a desired PLR and rocked overnight in a cold room. Next day, Bio-Beads were added stepwise to gradually remove detergents. After reconstitution, the vesicle solution became cloudy, and was aliquoted, flash-frozen in liquid nitrogen, and stored at −80°C until experimental use. Control vesicles were prepared similarly without protein. Other

proteins were treated in the same procedure, even though some of them were soluble and did not get into the vesicles at all.

For vesicle floatation assay, reconstituted vesicles were mixed with 20% Ficoll 400 by 1:3 volume ratio. The mixture (~0.3 ml) was the bottom layer, above which 10% and 5% Ficoll were loaded sequentially. The gradient was centrifuged at 250,000 $g$ for 3.0 h at 4°C. The vesicles migrated to the top 5% layer and were clearly visible as a narrow band in the gradient. Non reconstituted proteins stayed at the bottom. The gradients were fractionated for protein detection.

### Recordings of single-channel events in bilayer lipid membranes

The CHGB in egg PC vesicles at 0.50 mg/ml in varying PLR of 1:10,000 to 1:2,000 was first tested in 150 to 300-$\mu$m bilayer membranes. We observed that immediately after vesicle fusion the membranes were quiet and stable, but in some cases there were sudden changes in a short while (within a few minutes) with multiple high-conductance events, which usually became quiet again after a few tens of seconds. These observations have defied our efforts to record macroscopic currents from dozens or hundreds of CHGB channels.

Instead, vesicles with PLR of 25,000 to 1:10,000 led to small currents (usually <100 pA at 100 mV) from a handful of channels. The channels remained mostly open in low $Vm$ ([–50, +50 mv]), but frequently closed in higher voltages ($Vm > 80$ or $< –80$ mV). These channel events often occurred for tens of minutes and then disappeared. We suspected that certain lipid or solvent (decane) effects might cause such behavior in our recordings from lipid bilayers and will need to use a bSUM to study them (Zheng et al, 2016).

Vesicles of molar PLR 1:10,000 have on average ~2 CHGB tetramers per vesicle (assuming 100 nm diameter). Bilayers were prepared as before (Lee et al, 2013). Fusion of the CHGB vesicles with painted bilayer lipid membranes was induced by slow water flux from $trans$ to $cis$ side. After the appearance of channel activities, salt concentration was balanced. To achieve more reliable fusion of vesicles, 0.5 mM $CaCl_2$ and 1–2 mM $F^-$ ions were introduced in the $cis$ side. During the experiments, we eliminated those membranes showing a negative leak current greater than –1.0 pA at 0 mV before salt balancing in the $trans$ side because a small current more positive than –1.0 pA was expected from anion conduction, indicating a tight membrane with little leak. Voltage-clamp mode was used to recording channel activities.

An Axopatch 200B amplifier interfaced with a Windows PC through a DigiData 1322A analogue-to-digital converter was used for recordings. The Axoclamp software (Axon Instrument from Molecular Devices, Inc.) controlled experimental protocols. Clampfit (Axon Instrument) was used to measure currents and single channel events and further data analysis was performed in IGOR Pro (WaveMetrics, Inc.). Current recordings were filtered at 1.0 kHz using a Bessel filter and sampled at 2–5 kHz. The liquid junction potential between the solutions we used was <0.1 mV. With balanced salt solutions and under 0 mV holding potential, the recording system had a –0.8 to –1.0 pA leak current across a thinned membrane in the absence of any channels. This small negative current was corrected when reversal potential was read out from fitting the I–V curves.

For the recordings made from bilayers under asymmetric $Cl^-$ and symmetric $K^+$, the $cis$ side had 1.5 mM KCl, 150 mM K-isethionate, 10 mM MES-HCl, pH 5.5. The $trans$ side started with 15 mM KCl and 10 mM MES, pH 5.5, and was balanced with 135 mM K-isethionate after the appearance of channel activities. The liquid junction potential between these two solutions was measured to be close to zero. In these solutions, we recorded no significant outward currents.

We also tested the patching of blebbed membranes fused from reconstituted vesicles without much success. Recordings from giant unilamellar vesicles need a completely different setup, and will be tested in the future for a separate publication.

### Measurement of chloride efflux by an Ag/AgCl electrode

Direct $Cl^-$ efflux from vesicles was measured as described previously (Stockbridge et al, 2013). CHGB liposomes of 1:2,000 PLR (equivalent to ~10 CHGB tetramers per 100-nm vesicle) were loaded with 300 mM KCl and extruded through a 400-nm membrane filter and passed through a 1.5-ml G-50 Sephadex desalting column in a buffer containing 300 K-isethionate, 25 mM Hepes, pH 7.4, 1.0 mM EDTA, 2.0 mM $\beta$-ME, 0.2 mM KCl. Vesicles (~100 $\mu$l; inside KCl and outside K-isethionate) were added to the 1.0-ml recording chamber. The ground Ag/AgCl was connected to the recording chamber through a salt bridge. The recording Ag/AgCl electrode was directly immersed into the recording solution. ~0.2 mM KCl was added to the recording solution to stabilize the baseline. DMSO was added as control before valinomycin in DMSO stock was introduced at 0.25–1.0 $\mu$M to trigger the efflux of $Cl^-$. A stirring bar was used to mix valinomycin with vesicles well. Currents were recorded in the whole-cell mode under $V_m = 0$ mV and at the lowest gain. The data were filtered at 200 Hz and sampled at 2 kHz. Gap-free recordings were made for 45–60 s. At the end of the experiments, 50 mM $\beta$-octylglucoside ($\beta$-OG) was added to release all chloride.

### Measurement of chloride efflux from vesicles by the light scattering assay

For steady-state fluorimetry, we followed closely what was described previously (Stockbridge et al, 2013). Vesicles were loaded with 300–450 mM KCl and 20 mM Hepes at pH 7.4 and were extruded 20 times across a 400-nm membrane filter (Avanti Polar Lipids) before being desalted into a buffer containing 300 mM potassium isethionate. Liposomes were diluted to ~0.50 mg/ml lipids into 1.0 ml desalting buffer in a stirred cuvette. 1.0 $\mu$M valinomycin from 1.0 mM stock solution was added and mixed (~10–15 s) to start the flux. After ~30 s, light scattering was measured at 600 nm inside a Horiba fluoroLog spectrophotometer (HORIBA Scientific Inc.) using a Fluorlog-2 module at UT Southwestern (Jin et al, 1999; Stockbridge et al, 2013). This assay is called the light-scattering–based flux assay. At University of Florida (UF), it was repeated using a Fluoromax-4. Data analysis was performed in Origin.

To test the ion selectivity, CHGB proteoliposomes were prepared with different salts (e.g., KCl, KBr, KI, etc.). The proteoliposomes were extruded and desalted in the K-isethionate buffer and used to test the ion flux as described above.

To titrate the channels per vesicle, molar PLR varied from 1:100,000 to 1:1,000. Signals from three independent experiments (different batches of vesicles) were averaged and normalized against the maximal signals. Data were fitted with a Hill-equation as follows:

$$\Theta = \Theta_{max}\big/\big(1 + (k_D/[\text{CHGB}])^n\big) + C$$

The estimated Hill coefficient, $n$, for the steady-state measurement is ~1.4, suggesting that a functional channel needs more than one CHGB subunits. The steady-state measurement suffers from uneven mixing of valinomycin with vesicles because of uneven partitioning at the starting point and the slower water diffusion during the relaxation step after a change in vesicle shape.

### Preparation of bacterial EriC transporter

The expression construct was obtained from Dr. Christopher Miller at Brandeis University. The protein expression and purification followed a published procedure (Maduke et al, 1999). EriC protein was purified using Ni-NTA affinity chromatography and size-exclusion chromatography in a Superdex 200 column (GE Healthcare). Protein concentration was measured. Proteins were reconstituted in egg PC in the same way as described above for CHGB reconstitution.

### A stopped-flow system to observe the fast kinetics of anion flux

To overcome the shortcomings of the steady-state experiments, we modified the light-scattering based flux assay and used a stopped flow system to achieve quick mixing of equal volumes of valinomycin solution and vesicle solution. An Applied Photophysics SX20 MV stopped-flow spectrophotometer (dead time ~ 2 ms) in Dr. Linda Bloom's lab was used. It has an observation cell length of 1.6 cm and a mixing chamber of 110 $\mu$l. Right before each experiment, vesicles loaded with 300 mM KCl were exchanged into a buffer containing 300 mM K-isethionate and 10 mM Hepes, pH 7.4 and then loaded into one of the injection syringes. The other injection syringe contained 2.0 $\mu$M valinomycin in the same buffer. All visible air bubbles were carefully removed. Injection of 55 $\mu$l of solutions from both syringes started the experiments. The reactions were performed at 20°C with a water-bath controlling the temperature of the monitoring cell. Even mixing was achieved within 2 ms. The final mixed solution had 1.0 $\mu$M valinomycin, ~0.4 mg/ml lipids (egg PC), and a varying amount of CHGB protein (PLR). After mixing, light scattering at 610 nm was monitored. Each data point was the average of three consecutive scans, and each experiment was repeated three times using CHGB protein from different batches of purification and reconstitution. Titration of PLRs was performed in a continuous run on the same day to prevent system variation. Because of slow decay phase after the initial jump, we focused on the first 40 ms in our analysis. To titrate valinomycin, CHGB vesicles of PLR = 1:2,500 was used at varying concentrations of valinomycin. Fitting with a Hill-equation yielded a Hill coefficient of ~2.6, suggesting strong cooperativity for valinomycin in transporting K$^+$ ions to support the fast flow of Cl$^-$ ions through CHGB. The IC50 = 650 nM, which is approximately 130 valinomycin in each 100-nm vesicle.

### Negative-stain electron microscopy (EM) of reconstituted CHGB vesicles

Copper grids coated with a thin layer of carbon film were baked at 70°C overnight the day before experiments. After glow discharge of the grid, 3 $\mu$l of reconstituted CHGB vesicles with specific PLRs were loaded. The sample was incubated on the carbon-coated grid at RT for 30 s before being blotted. After that, the grid was stained with 2.0% phosphotungstic acid (PTA), pH ~8.0 for 30 s. The grid was blotted and air-dried before being observed in a JEOL JEM2200FS microscope. Images were taken with a Gatan K2 Direct Electron Detector at 25,000× with a defocus level of −2.0 $\mu$m and a calibrated pixel size of 1.92 Å.

For 3D reconstruction from negative-stain EM images, images were recorded with an electron dose of 20 e$^-$/Å$^2$ on a 4K × 4K Gatan K2 Summit Direct Electron Detector (Gatan) in a counting mode. 180 images were selected based on the power spectra determined by CTFFIND3 (Mindell & Grigorieff, 2003). 140 images with minimal astigmatism were selected for particle picking. The particles were selected using the Boxer module in EMAN 2 (Tang et al, 2007). A total of ~5,400 particle images were manually selected. An initial model was generated by angular reconstitution in IMAGIC 5 (van Heel et al, 1996) and finally refined in SPIDER (Frank et al, 1996; Jiang et al, 2004). The final map was calculated from ~3,000 particle images at a nominal resolution of 30 Å. The handedness of the map was tested with paired images taken from the same specimens at both +15 and −15° as what was carried out for the C3PO negative stain map (Llaguno et al, 2014). The small, compact size of the CHGB dimer made us less confident in the handedness at this point. A high-resolution map will be needed to further examine chirality.

### CryoEM study of CHGB dimers in detergents

Quantifoil R2/2 grids (Quantifoil Micro Tools GmbH) were coated with a thin carbon film (~2–3 nm). The ChemiC (Ni-NTA) grids were prepared as described before (Llaguno et al, 2014). 3.0 $\mu$l of purified CHGB in detergents was loaded. After incubation for ~15 min in a wet-chamber of >90% humidity, the grid was blotted inside a Vitrobot and plunge-frozen into liquid ethane bathed in liquid nitrogen (FEI). After screening in a JOEL2200, good specimens were imaged at HHMI Janelia Farm Research Campus. A Titan Krios microscope equipped with a 4K × 4K Falcon 2 Director (no movie function at the time) was used. The scope was operated at 300 kV and was equipped with a Cs corrector. Automatic data collection was run by a proprietary software package, EPU (FEI). Images were taken under a defocus within a range of −2.5 to −4.0 $\mu$m at a magnification of 37,000×, which gave rise to a calibrated pixel size of 1.89 Å at the specimen level.

Because the CHGB dimers were quite small, we scanned many areas for good recognition of the particles. Only a small dataset was successfully built, which came from ~300 of 4K × 4K images. These images all displayed good Thon rings to a resolution of ~ 6.0 Å with minimal astigmatism and defocus values ranging from −1.0 to −4.0 $\mu$m, and showed visible particles. 24,086 particles were picked manually and extracted in 196 × 196 Å$^2$ boxes. The low-resolution negative-stain map was used as the reference for 3D refinement. Five rounds of 2D classification into 50 distinct classes were performed with the program RELION 1.3 (Scheres, 2012). The classes with

well-defined particles were selected. 3D classification was performed with these selected particles into five classes. Two classes showing higher resolution features were selected for further refinement. The 12,123 particles that were assigned to these two classes were subjected to five additional rounds of refinement using a high-resolution frequency limit of 6 Å. A soft mask was introduced to redo the 3D classification and remove ~45% of the particles. The final map was calculated from ~6,900 particles and the estimated resolution was 9.8 Å by using a threshold of 0.143 to the gold-standard Fourier shell correlation (Rosenthal & Henderson, 2003; van Heel & Schatz, 2005). The map was sharpened by applying a negative B-factor of −75 Å$^2$.

## Measuring Ca$^{2+}$ efflux from reconstituted vesicles

Proteins (CHGB, its mutants, or IP$_3$R) were reconstituted into vesicles in a PLR of ~1:5,000 in a buffer made of 20 mM Hepes, pH 7.5, 100 mM NaCl, 1.0 mM EDTA, and 2.0 mM β-ME (high pH) or a buffer made of 20 mM MES, pH 5.5 (low pH). To load CaCl$_2$, 1.0 mM CaCl$_2$ was added to the buffer. Right before each experiment, freshly prepared lipid vesicles (10 mg/ml) was extruded 20 times through a membrane filter with an average pore size of 100 nm (Avanti polar lipids). A PD-10 desalting column (Sigma-Aldrich) was used to change the vesicles into a buffer containing 100 mM NaCl and 20 mM Hepes, pH 7.4, 1.0 mM EDTA and 2.0 mM β-ME. Chelex 100 was used to treat the buffers to remove residual calcium ions. Vesicles coming out of the column (~3.0 mg/ml lipids) were diluted to 0.2 mg/ml into the external buffer inside a quartz cuvette with constant stirring. A Ca$^{2+}$-sensitive fluorophore, Indo-1, was added to 1.0 μM. The efflux, if any, was initiated by adding 0.5 μM valinomycin. Indo-1 fluorescence at 410 nm was measured in a Horiba FluoroLog spectrophotometer (HORIBA Scientific Inc.) using the Fluorolog-2 module and an excitation wavelength of 330 nm.

For the IP$_3$R-containing vesicles, the protein was purified from rat cerebellum as reported before (Jiang et al, 2002), and reconstituted in egg PC in the presence of 2.0 mM CaCl$_2$. The efflux of calcium was initiated by adding 1.0 μM IP$_3$ at different time points. At the end of the experiments, 30 mM β-OG was added to disrupt the liposomes and release all calcium ions to determine the maximum signal.

## Mass spectrometric and Proteomic analysis of proteins

Trypsinization and mass spectrometry analysis of the protein bands were performed by the Protein and Peptide core facility at UT Southwestern Medical Center. The N-terminal sequencing of the purified fragments eluted from the SDS–PAGE was performed by the same facility. Standard procedures at the core facility were used. After my lab was relocated, the Proteomics core at UF Interdisciplinary Center for Biotechnology Research (ICBR) performed the same services.

The excised bands from the Coomassie-blue–stained SDS–PAGE gels were submitted for identification. The gel bands were destained with 1.0 ml of 50 mM ammonium bicarbonate, pH 8.0/acetonitrile (1:1, vol/vol). Each sample was reduced with 40 mM DTT, alkylated with 100 mM of iodoacetamide, and trypsin-digested. Trypsin-digested peptides were desalted with C18-Ziptip (Merck Millipore). A hybrid quadrupole Orbitrap (Q Exactive Plus) MS system (Thermo Fisher Scientific) interfaced with an automated Easy-nLC 1200 system (Thermo Fisher Scientific) was used. Samples were loaded into an Acclaim Pepmap 100 pre-column (20 mm × 75 μm; 3 μm-C18) and separated in a PepMap RSLC analytical column (250 mm × 75 μm; 2 μm-C18) with a flow rate of 250 nl/min using a linear gradient from solvent A (0.1% formic acid [vol/vol]) to 25% solvent B (0.1% formic acid [vol/vol], 19.9% water [vol/vol], and 80 % acetonitrile [vol/vol]) for 80 min, and then to 100% solvent B for additional 15 min.

The candidates were of 97.0% probability and contained at least 1 identified peptide (Keller, et al, 2002). Protein probabilities were assigned by the Protein Prophet algorithm (Nesvizhskii et al, 2003). Proteins that contained similar peptides and could not be differentiated based on MS/MS analysis alone were grouped to satisfy the principles of parsimony. Proteins sharing identified peptides were grouped into clusters.

Four shared peptides between the two contaminating bands suggest that they are of the same origin. All other candidates only had one peptide detected, very unlikely to be real candidates. Among all the contaminants there is not a single one that can serve as an anion channel.

## Ion flow through the CHGB requires a fast-conducting channel, not a slow-acting transporter

The main difference between a transporter and a channel is the flux rate. Even for the fastest known ion transporter (Cl$^-$/HCO$_3^-$ transporter), its turnover rate (up to 10$^5$ per second) would still be a couple of orders of magnitude slower than that (~10$^7$ per second) of a channel. The stopped-flow–based flux assays can provide a good estimate of the flux rate that is limited by the maximum flow through valinomycin molecules. Titration of valinomycin concentration in Fig 6H suggested that ~200 valinomycin molecules per vesicle were needed to generate a significant signal within the 2-ms period of quick mixing. We assume that almost all valinomycin molecules partition to lipid membranes because of their hydrophobicity. Valinomycin shows a turnover rate of ~10$^4$ per second at RT in the absence of a transmembrane electrostatic potential. Thus, given the parameters for individual CHGB vesicles (White and King, 1985; Table S3), the estimated K$^+$ flux per 100 nm vesicle is ~2 × 10$^6$ per second, or ~4 × 10$^3$ in 2 ms, which alone is not enough to release a significant fraction of ~10$^5$ K$^+$ ions from each 100-nm vesicle without a transmembrane potential.

In the beginning of changing the vesicles from 300 mM KCl into 300 K-isethionate, initially, the Nernst potential for Cl$^-$ was infinitely positive. The mixture had a lipid concentration of ~0.4 mg/ml, a PLR varying from 1:100,000 to 1:1,000 (molar ratio), and 1.0 μM valinomycin. After a small leak of Cl$^-$ ($W$, the number of ions leaked out) to the outside, an electrostatic potential was established. The Nernst equation was related to the charging of the vesicles to determine the number $W$.

$$-\frac{RT}{F}\ln\left([vesicles] \times W/[Cl]_i\right) = W \times e_0/C_{vesicle}$$

$R$, gas constant; $T$, temperature; $F$, the Faraday's constant; $[vesicles]$, average concentration of vesicles; $e_0$, elementary charge; $C_{vesicle}$, capacitance of one vesicle of average size (100 nm).

A numerical solution of $W$ was obtained using MATLAB (Table S3).

It was noted that the Nernst potential became stable once decreased from the initial infinity. From these estimates, a strong Nernst potential inside would drive the valinomycin transport of $K^+$ ions out. An electrostatic driving force for $K^+$ efflux at the first 2 ms can be expressed as a correction factor, $exp(z\delta FV/RT)$, which is ~$8.9 \times 10^4$ at +295 mV, where $\delta$ is a factor for effective conversion of the electrostatic energy into the driving force for ion movement, and is assumed to be unity here under the consideration of no significant energy loss due to either partial charge loss or charge delocalization during the movement of the $K^+$/valinomycin across the bilayer. The correction factor would increase the initial $K^+$ flux rate through ~200 valinomycin molecule in each vesicle to as high as ~$1.8 \times 10^8$ per ms at +295 mV, which is the peak rate. Once the $K^+$ and $Cl^-$ ions start to flow out, the correction factor quickly decays when the vesicular potential drops to below +180 mV (Fig 5H). Such a flux rate would be enough to quickly dump most of the ~$10^5$ $K^+$ ions inside each vesicle in <1 ms, and cause a significant, sudden decrease in osmolality and thus a sudden collapse of the vesicles before water diffusion could follow the shape change of the vesicles. The water movement was found to be in the tens of ms range. This explanation would also account for the sensitivity of the light-scattering signal to [valinomycin]. Moreover, for the fast removal of KCl from the interior of each vesicle, the efflux of chloride ions had to match that of the potassium ions. A lower limit is ~$10^{7-8}$ per second. Possible movement of water molecules accompanying the movement of $Cl^-$ only accounts for a tiny fraction of water molecules inside each vesicle.

Our calculation made a testable prediction that a slight increase in [$Cl^-$] outside of the vesicles would diminish the flux signal by influencing the correction factor. Our data showed that 2–5 mM KCl in the extravesicular side almost completely abolished the light-scattering signal. It means that when the initial transmembrane Nernst potential drops to <+150 mV inside (0.875 mM KCl outside), the correction factor of $exp(z\delta FV/RT)$ decreases by ~290-fold such that the average KCl efflux rate would become significantly slower than what is needed to generate a sudden drop in osmolality. Our experimental data confirmed this prediction.

Based on the titration of PLRs in CHGB vesicles in the stopped-flow assay, the average number of CHGB subunits per vesicle needs to be at least four per vesicle for generating a fast signal, which is likely equivalent to one channel, and 6–8 CHGB subunits per vesicle are sufficient to reach a maximum signal because of the random Poisson distribution (see the next section). That is, on average, 1–2 CHGB tetramers are sufficient to achieve the fastest efflux estimated. A flux rate of ~$10^5$ ions per ms per molecule leads to an estimated single-channel conductance of ~100 pS in the presence of 300 mM KCl, close to the measured value.

The fast mixing and even partitioning of valinomycin with the vesicle suspension in the stopped-flow mixing cell and the relatively lipophilic nature of valinomycin makes it possible for us to assume that it takes no free energy for valinomycin to diffuse through the aqueous phase in an estimate velocity of ~1–2 $\mu m$/ms via random walk, and become enriched in bilayer membranes. The efflux of a small amount of $K^+$ ions carried by valinomycin is followed by the efflux of $Cl^-$ ions through the CHGB channels such that the positive potential inside the vesicles would fall quickly from the starting +295 mV, following the exponential curve. The

change in potential will slow down the efflux of $K^+$/$Cl^-$ and eventually stop further change in osmolality and vesicle shape.

### A Poisson distribution of CHGB subunits among vesicles predicts tetrameric stoichiometry of functional channels

In the description of the Poisson distribution in the main text, we considered the following assumptions. (1) When M monomers are available in a vesicle and sufficient to join together to form a channel, the assembly is nearly 100% efficient. Based on the biochemical data, the CHGB protein was fairly stable and our reconstitution was able to incorporate all proteins into vesicles in a preferred orientation; (2) One channel suffices to conduct enough $Cl^-$ within 1–2 ms. The conductance of CHGB with 300 mM $Cl^-$ would be >125 pS; (3) With assumption one, all vesicles with less than M copies of CHGB monomers will have no channel and will not contribute to the change in light-scattering. They will form a non functional fraction with a probability of $P$ ($k < M$); (4) Vesicles with more than M monomers (N > M) will form at least one functional channel (using M subunits), and the surplus monomers (N–M; very likely dimers as the basic units) will not interfere with the function of the assembled channels. With a large number of vesicles for our experiments, the statistical average will likely overcome small experimental variations in vesicle size, completeness in the CHGB insertion into individual vesicles, freedom of CHGB dimers to diffuse and interact with each other in vesicles to form channels, ratios of lipid types in the egg PC mixture, diffusion time for valinomycin molecules to arrive at membranes, and degrees of reaching complete mixing of two equal volumes within the 2 ms dead time. Therefore, it is relatively reasonable to consider that these assumptions will be satisfied.

Least squares calculation found that the best fit is M = 4, which is better than M = 5, but much better than M = 3, 6, 7, or 8. Because of the stability of CHGB dimers, M = 5 is not a good solution. The agreement of the predictions from Poisson statistics and the experimental data appears to not be coincidental, especially at the lower range of average CHGB monomer per vesicle ($\lambda \leq 4$), where we expect that the assumptions of random distribution and efficient assembly of functional tetrameric channels are well satisfied. Considering the dimers being the dominant species in detergents and the tetrameric form observed in the chemically cross-linked fractions, our data support the tetramers as functional channels, not the pentamers, trimers, or other oligomers.

## Supplementary Information

## Acknowledgements

We are grateful to Dr. Barbara Ehrlich (Yale University) for providing the construct for mCHGB, to Dr. Christopher Miller (Brandeis University) for sharing the construct for EriC and the protocols for the chloride flux assays, to Dr. Herbert Y Gaisano (University of Toronto) for the syncollin-pHluorin construct, to Dr. Kuixing Zhang (UC San Diego) for sharing his techniques for releasing granular contents from endocrine cells, to Dr. Sohini Mukherjee and Lora Hooper (UT

Southwestern Medical Center) for letting us use their FluoroMax-3 system for studying the release of fluorophores from reconstituted vesicles, and to Dr. Sandra Schmid (UT Southwestern Medical Center) for accessing a Horiba fluorometer. We thank Drs. Jose Rizo-Rey, Paul Blount, Peter Michaely, and Lily Huang at UT Southwestern, Frederick Sigworth at Yale University, Drs. Zhonglin Mou and Julie Furlow-Maupin at the University of Florida, and Michael X. Zhu at UT Health Science Center at Houston for critically reading and commenting on the manuscripts, and Dr. Christopher Miller and Dr. Randy Stockbridge for in-sightful discussions on the flux and bilayer data and the interpretation of the results. We would like to express special gratitude to Drs Jin Koh and Sixue Chen in the Proteomics Core at the Interdisciplinary Center for Bio-technology Research (ICBR) of the University of Florida for technical support in mass spectrometry and proteomic analysis. The work was mainly supported by National Institutes of Health (NIH) (R01GM111367 and R01GM093271 to Q-X Jiang), CysticFibrosis Foundation (JIANG15G0 to Q-X Jiang), Welch Foundation (I-1684 to Q-X Jiang), and Cancer Prevention and Research Institute of Texas(RP120474 to Q-X Jiang), and partially by an American Heart Association National Innovative Award (12IRG9400019 to Q-X Jiang), an National Institute of General Medical Sciences (NIGMS) EUREKA Award (R01GM088745 to Q-X Jiang), and the startup funds from the University of Florida. Some of the experiments reported here were performed in a laboratory constructed with support from NIH grant C06RR30414 with Dr. Jerry Shay as the PI. Data collection of cryoEM specimens was performed at Howard Hughes Medical Institute Janelia Farm Research Campus and at the Florida State University-based "Southeast Consortium for Microscopy of MacroMolecular Machines" supported by an NIGMS grant U24GM116788 (Dr. K Taylor as PI and Q-X Jiang as one of the co-PIs).

## Author Contributions

G Yadav: data curation, formal analysis, investigation, methodology, and writing—original draft, review, and editing.
H Zheng: data curation, formal analysis, and writing—review and editing.
Q Yang: data curation, methodology, and writing—review and editing.
L Douma: data curation, formal analysis, methodology, and writing—review and editing.
LB Bloom: data curation, formal analysis, supervision, and writing—review and editing.
QX Jiang: conceptualization, resources, data curation, software, formal analysis, supervision, funding acquisition, validation, investigation, visualization, methodology, project administration, and writing—original draft, review, and editing.

## Conflict of Interest Statement

The authors declare that they have no conflict of interest.

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
