## [Reviewer comments · Life Science Alliance]

Secretory granule protein chromogranin B (CHGB) forms an anion channel in membrane

Gaya P. Yadav, Hui Zheng, Qing Yang, Lauren G. Douma, Linda B. Bloom and Qiu-Xing Jiang DOI: 10.26508/lsa.201800139

Review timeline:

Submission Date:	26 July 2018
Revision Received:	26 July 2018
Editorial Decision:	30 July 2018
Revision Received:	29 August 2018
Accepted:	11 September 2018

Report:

(Note: Letters and reports are not edited. The original formatting of letters and referee reports may not be reflected in this compilation.)

Please note that the manuscript was previously reviewed at another journal and the reports were taken into account in inviting a revision for publication at *Life Science Alliance* prior to submission to *Life Science Alliance*.

1st Revision – authors' response

26 July 2018

Referee #1: Major points:

1. *Any chloride channel I have heard about has permeability for Br⁻, which is almost as high - or higher - than Cl⁻. This is because Br⁻ has about the same size as Cl⁻ and almost the same total energy of hydration. The permeability to F⁻ is almost always lower than Cl⁻, because the energy of hydration is higher. Thus, the permeability sequences the authors determine seem to be inconsistent with chloride channels as we know them.*

Our experimental data have been highly reproducible when we repeated multiple times the same types of experiments using proteoliposomes prepared from different batches of protein preparations. We believe that the CHGB channel is unique because all other five well-known families of Cl⁻ channels are canonical membrane proteins with transmembrane alpha-helices. Instead, the CHGB channel is formed through amphipathic helices. In this sense, it is not surprising that our observed anion selectivity of CHGB channel differs from that of other Cl⁻ channels in literature.

Chemically, Br⁻ and Cl⁻ have diameters of 1.87Å and 1.67Å, respectively, and different thermodynamic terms: entropy of 82 vs 153.4 J/mol/K; and enthalpy of -121 vs -167 kJ/mol. How the CHGB channels can distinguish them will need atomic resolution structures of the CHGB channel soaked with either Br⁻ or Cl⁻. If the permeation pathway in the channel allows hydrated ions to pass, the thermodynamic terms may be sufficient to generate the difference in flux.

During the revision period, we identified a few point-mutations that affect the ion-flux (Fig. 5E in paper1). As a different example, the Fluc F⁻ channel (Stockbridge, et al, eLife, 2:e01084, 2013) is quite unique, as it conducts F⁻ much better than Cl⁻ or other anions. CHGB conducts F⁻ and Cl⁻ equally well. Further, we performed ion competition assays between Br⁻ and Cl⁻, between I⁻ and Cl⁻, or between F⁻ and Cl⁻, and the data agreed with the measured selectivity sequence (Paper 1, Fig. 5). We introduced DIDS to inhibit the CHGB channel. The apparent affinity for DIDS is much higher for CHGB (0.5 microM) than for other Cl⁻ channels (for example, 300 micromolar for ClC-ec1). We therefore propose that the CHGB family forms a sixth family of Cl⁻ channels.

2. *The channels in the black lipid membrane have not been properly characterized. What are their selectivity to different anions - and what is the permeability ratio? In Fig. 5D, why is the reversal potential found by extrapolation from data obtained < -60 mV? Clearly, transitions can be measured all the way up to 1 pA. Did the authors actually ever observe transitions at 0 mV difference between chambers? In Fig. 5B, why are there no points on the curve between 0 and +60 mV? These might be transitions through intermittent lipid channels, possibly induced by amphipathic helixes, but with no ion selectivity.*

We did more experiments to address reviewers concern. The permeation ratio between F- and Cl- is ~ 1.2 (Fig. 4F in paper 1), consistent with the flux assay data (Fig. 5F in paper 1). We have also added more data points in the figure 4B of paper 1, which was the Fig. 5B in the last submission. In figure 4C/4D of paper 1, the reversal potential for Cl- is +60 mV, and the reversal potential by extrapolation is +65 mV. We did extrapolation mainly because we were not able to record any current at the voltage-range of -40 to +80 mV. Due to the ionic conditions, the current close to 0 mV is quite small ($0.065 \text{ V} * 58 \text{ pS} \sim 3.8 \text{ pA}$) for bilayer recordings in our system. There is no current for positive potential because of low Cl- (1.5 mM) in the cis side.

3. *Results: the fact that the authors needed Triton-100 to solubilize the protein is not evidence that it is membrane associated.*

Yes, it is true. We could not claim CHGB as a membrane-associated protein solely based on the need of Triton X-100 for its stability during purification. We established CHGB as a membrane-associated protein by floatation experiment of the CHGB vesicles made from recombinant rat CHGB and native bovine CHGB (Fig. 1E in paper 1 and Fig 2C in paper 2). Both soluble-fraction and membrane-bound fraction from purified pancreatic granules contain CHGB proteins that could be reconstituted into membranes and form ion channels (Fig. 1D, 1F, 2D in paper 2). We did a new set of experiments to label CHGB on the INS-1 cell surface after granule release (Fig. 3A-C in paper 2), which is consistent with the published data from PC-12 cells (Pimplikar and Huttner, 1992). All these data now together strengthen our conclusion that the CHGB can be membrane-inserted and form an ion channel.

We deduced the molecular size of the CHGB using size-exclusion chromatography. On the basis of the molecular weight standards CHGB, equivalent to a 300 kDa globular protein, is likely a trimer or a dimer because each monomer is ~100 kDa. The average micellar size of Triton X-100 is ~100 kDa, suggesting that the CHGB mass in detergents is likely 200 kDa in each micelle, thus a dimer. Due to the inaccuracy in size-exclusion and the shape of the proteins, we further confirmed the dimeric nature by images analysis and 3D reconstruction (Fig. 1C, 1D in paper 1).

4. *Results: "Due to the detergent micelle (~100 kDa), the CHGB was likely a dimer, instead of a trimer." What does this mean?*

We deduced the molecular size of the CHGB using size-exclusion chromatography. On the basis of the molecular weight standards CHGB, equivalent to a 300 kDa globular protein, is likely a trimer or a dimer because each monomer is ~100 kDa. The average micellar size of Triton X-100 is ~100 kDa, suggesting that the CHGB mass in detergents is likely 200 kDa in each micelle, thus a dimer. Due to the inaccuracy in size-exclusion and the shape of the proteins, we further confirmed the dimeric nature by images analysis and 3D reconstruction (Fig. 1C, 1D in paper 1).

5. *The data using 'a 10 Å fluorescent dye' (which one?), which was not released by valinomycin, should be shown.*

Such data are presented in Fig 3A in paper 1.

As negative control, vesicles without CHGB protein were added to Figure 3C in paper 1.

6. *Figure 3A-B. A control experiment showing what happens in the absence of CHGB (negative control) is missing. Also, a control experiment showing what happens in the presence of a verified Cl-channel would be advisable (positive control).*

We have not done direct current measurement for a known chloride channel as a positive control. We do have data for the anion flux through EriC in Fig. S4H in paper 1.

7. *The assay making use of vesicles turning into ellipsoids (Fig. 3C) is new for me: has this been verified and used by others? It takes more than the presented data to verify this assay. What is the evidence that vesicles are more ellipsoid when the scattering is high?*

This method was developed by the Christopher Miller's lab, and was used by them and other labs very often. The fast efflux of KCl led to a sudden decrease of osmolality and the collapse of the spherical vesicles, which could be detected as a change in light scattering. The light scattering increases with the increased Stokes radii of vesicles, which happens when the vesicles collapse from a spherical shape into a disc-like or oblong shape. Please refer to the following paper.

References:

- Stockbridge RB, Robertson JL, Kolmakova-Partensky L, & Miller C (2013) A family of fluoridespecific ion channels with dual-topology architecture. *Elife* 2:e01084.

8. *The authors misquote the scientific literature and twist the record to suit their own ends:*
A. *In the Introduction, the authors seem to construct a need of a Cl-conductance for 'maturation of nascent immature secretory granules', by citing two papers from 1978 and 1979: Brown et al., 1978 (ref.8) and Johnson and Scarpa 1979 (ref. 9). The first thinks of exocytosis as something similar to osmotic rupture, which is clearly no longer valid. The second only concerns isolated chromaffin granules, and has nothing to do with chloride. If anything, this publication shows that thiocyanate can collapse the membrane potential of granules, which argues against the present findings of a channel, which is not permeable to thiocyanate. For both of the publications, there is no appreciation of immature secretory granules, which were not studied in the 1970s.*

B. *The authors state that the "CLC-3 transporter was once proposed as a candidate, but its granular localization was ultimately negative (10-13)". However, two of the four papers state that CLC-3 is in the granule. One paper (#12) states that it is not, and then there is #11, which is a comment by Ueno, wrongly attributed to Jentsch, and later retracted and republished under Ueno's name, with an answer by Jentsch. However, it has nothing to do with CLC-3, but CLC-2. Probably, this (retracted) comment is not what the authors mean to cite, but a 2010-comment by Jentsch in *Cell Metabolism* on the localization of CLC-3. So, according to these papers, two groups think it is on the granule, and one thinks it is not - how does that justify the author's statement?*

The literature was cited with the intention of showing that the questions regarding the Cl- channels in different processes are not fully resolved. We reorganized our thoughts and citations to discuss the different anion channels that might function in different steps in paper 2: Cl- influx during granule maturation, Cl- efflux before granule release, Cl- influx at plasma membrane before Ca²⁺-triggered exocytosis, Cl- flux during early endocytosis and possible Cl- flux in late endosome and its fusion /sorting to TGN or ISGs, etc. For the Cl- conductance at the plasma membrane during exocytosis, the Ca²⁺ influx into the cells for exocytosis might be accompanied by a Cl- influx to maintain charge neutrality. The Johnson & Scarpa paper is related to ion permeation of chromaffin granules and connected to their other papers in early 1980s (Johnson et al 1982), now added at the introduction (line 61 in paper 1)

Thanks for pointing out the confusion between #11 and the Jentsch paper . We have corrected it. The antibody quality is the main cause for the discrepancies among the three papers regarding the presence or absence of CLC-3 in granules. Jentsch et al produced the antibodies and thoroughly characterized them. Their data are probably more reliable than the other two.

C. *The authors state that "CLC3's fS conductance and un-measurable outward anion flux make it unfit to carry a substantial Cl- flux into the secretory granule". However, most people now believe CLC-3 is a Chloride/proton exchanger, which would make it perfect for the job. Why do the authors not mention this?*

We moved this part to Discussions section in paper 1. There are still some people believing the CLC-3 can be a channel. The CLC-3, when delivered to cell surface (Guzman et al 2014; 2013), had zero

inward current, which corresponds to Cl⁻ efflux, suggesting that the flux into granules might be too tiny to support efficient acidification. Time-integral of slow Cl⁻ influx may not be sufficient due to the slow response rate. We leave this point open and so is the connection between CHGB and CIC-3. Further we would like to point out that the intragranular pH measurements by Deriy et al 2009 were not based on a large number of granules, which might need to be verified due to the distribution of measured pH (Fig. 4B, 4C in paper 1).

D. The author's state in the Introduction: "However, CHGB was still thought inaccurately to be soluble." Are the authors already concluding on their own study in the Introduction?

Thanks to the reviewer for pointing out this. We have rewritten the introduction and incorporated the two different forms of CHGB, from the soluble and membrane-bound fractions released from pancreatic granules, which were observed in the native bovine CHGB (Figs 1-2 in paper 2).

Minor:

Fig. 3D, E, F, and elsewhere: Remove white bars from traces, which obscure data at the time of addition of compounds.

These white bars were used to show the waiting time during the assay. We have removed them from all the figures wherever applied.

Referee #2:

1. Most of the data supporting that CHGB is a membrane protein comes from reconstitution using purified proteins. The authors should determine the localization of the native protein, or at least heterologously-expressed protein in its cellular environment. One, for example, can use a tagged CHGB and immuno-EM to determine whether the protein is in the granule membrane.

Instead of performing immuno-EM experiments, which require specific techniques not available to us, we did two alternative experiments. First, we found that native CHGB in INS-1 cells is on the cell surface after exocytotic release of the granules (Fig. 3A, 3D in paper 2). Both KCl depolarization and glucose treatment showed the same.

Yoo SH's group showed that CHGB is localized on the granular membrane of human astrocytes. Astrocytes from human brain tissues were immunolabeled for CHGB in figure 2A of (Hur et al 2010) with 15 nm gold with affinity purified CHGB antibodies and CHGB-labeled gold particles. The gold particles are very close to the granular membrane.

References:

Hur YS, Kim KD, Paek SH, Yoo SH (2010) Evidence for the Existence of Secretory Granule (Dense-Core Vesicle)-Based Inositol 1, 4, 5-Trisphosphate-Dependent Ca²⁺ Signaling System in Astrocytes. PLoS ONE 5(8): e11973.

Wolfgang Hartschuh, Eberhard Weihe and Ursula Egner (1990) Electron microscopic immunogold cytochemistry reveals chromogranin A confined to secretory granules of porcine Merkel cells. Neuroscience Letters, 116 (1990) 245 249.

Second, as advised by the reviewer, we moved to native CHGB to test the two different forms of CHGB reported literature. With the native bovine CHGB released from pancreatic granules, we partially purified CHGB from both soluble and membrane-bound fractions and found that both have the capacity of forming ion channels (Fig. 1 & 2 in paper 2), which means that when there is plenty of membrane area, most CHGB would be membrane-bound, but it can also exist as soluble fractions as nanoparticles (Fig. 2 in paper 1 or as complexes with other proteins or lipids (Fig. 2 in paper 2). It is probably attributable to the amphipathic nature of the CHGB membrane-interacting fragments.

2) The most direct evidence supporting that CHGB is an ion channel comes from single channel recordings using lipid bilayers (BLM, Fig. 5). BLM single channel recording is notoriously known to be prone to artifacts, particularly when used to define novel ion channels because of its high

sensitivity (it requires only one functional protein molecule to see a channel.) Although the authors did follow the general precautions, including the demonstration of protein purity and the inclusion of controls of truncated protein purified under identical conditions, more detailed characterization seems to be required. One way to increase the number of channels recorded is to record macroscopic currents from e.g. whole vesicles. At minimum, the authors should have a detailed comparison between the single channels recorded in lipid bilayers (Fig. 5) and the macroscopic conductance assayed using ion flux and light scattering (Fig. 3). Similar properties such as blocker sensitivity, ionic selectivity (Fig. 3G & Fig. 5E) between the two can give more confidence that the channels recorded in lipid bilayers are indeed formed by CHGB. The most convincing evidence can be from single point mutations that change channel selectivity, but those experiments are perhaps beyond the scope of the current paper.

We did use the ion flux data from vesicles to estimate a lower limit of single channel conductance (> 100 pS; see section 13 in supplementary information for paper 1). When we deleted the first half of the loop between helix 2 and helix 3 (Fig. 5E in paper 1; CHGB Δ L1 mutant), the channel function of CHGB was significantly impaired when compared with the same amount of wild-type CHGB protein reconstituted in parallel as control. This deletion mutation also narrows down a region more specific for ion conduction. Three point mutations 545A, 552A and 558A, were introduced and their effects on ion flux were weaker than CHGB Δ L1 (Fig. 5E in paper 1). We performed DIDS inhibition on the native and recombinant channels (Fig. 5B in paper 1 and Fig. 1G in paper 2) and found similar binding affinity. We measured the permeation ratio between F- and Cl- in lipid bilayers, which agrees well with the ion-flux assay (Fig. 4F vs. Fig. 5F in paper 1). We also performed mass spectroscopy analysis on the purified CHGB protein to rule out the possibility of the contaminants in the purified protein to be the ion channels (less than 0.2% in mass). All these together greatly enhance our confidence level.

Minor comments:

3) *The manuscript can be reorganized and shortened by, for example, moving part of Introduction to Discussion. There are also plenty of "loose" statements throughout the manuscript (some listed below); they require some careful editing.*

We have reorganized the introduction and discussions into the two manuscripts.

4) *page 4, "CIC3's fS conductance and un-measurable outward anion flux make it unfit to carry a substantial Cl⁻ flux into the secretory granules (14, 15)". CIC3's single channel conductance is actually >10 pS. In addition, many of the CICs actually have measureable inward currents (outward Cl⁻ flowing) even though they outwardly rectify (see e.g. Ishida et al. (2013) JGP 141: 705-720). There are two groups of results. The duality of CIC-3 as transporter and channel is still believed by some people in the field. In the papers by Guzman et al (2014) (2013), the CIC-3 Cl⁻/H⁺ exchanger was found to have very strict outward rectification. Because of these arguments, we moved the CLC-3 related points to Discussions section in paper 1.*

5) *page 5, "we found that CHGB in membrane formed a highly chloride-selective channel (29)." Reference 29 should not be cited here.*

Reference was removed from the reference list.

6) *page 6, "we realized that we could not treat the CHGB as a soluble protein. Instead, Triton X-100-like detergents were needed to prevent complete protein loss, meaning that the full-length CHGB is membrane-associated." I'd change "meaning" to "suggesting".*

Changes were made as suggested by the reviewer.

7) *page 8, "Calcium binding is a hallmark of CHGB." Please cite a reference.*

The reference was cited in the reference list and was also mentioned in the text at proper place.

• SH Yoo (1995) pH- and Ca²⁺-induced Conformational Change and Aggregation of Chromogranin B. COMPARISON WITH CHROMOGRANIN A AND IMPLICATION IN SECRETORY VESICLE BIOGENESIS. *The Journal of Biological Chemistry*, 270, 21, 12578- 12583.

8) page 11, subtitle "*CHGB can release interior chloride ions and is a Cl⁻ channel or transporter*". If the single channels recorded in Fig. 5 are formed by CHGB, then CHGB is very likely a channel because of its large conductance of ~60pS. I'd delete "or transporter".

The correction was made in the manuscript as suggested.

9) page 12, "..., we estimated that >70% of CHGB vesicles must have had Cl⁻ conductance in them". The efflux data can be used to estimate the percentage of enclosed volume but not the number of vesicles. For example, you can have half (50%) of the vesicles with much larger volume than the other half; release from the larger ones alone can account >70% of the total release.

We used an extruder to make vesicles more uniform in size to overcome the issues raised here. Further the flux assay analyzed ~10E11 vesicles, even 50% vesicles would allow us to test a huge number of CHGB channels in them.

10) page 12, "...CHGB must not pass K⁺ because valinomycin was required to elicit the Cl⁻-flux signal. It is therefore anion-selective." It's not clear to me whether one can conclude from the efflux measurement alone that the vesicle is not permeable to K⁺. For example, the vesicles can have basal K⁺ permeability (not reflexed in this assay) but valinomycin drastically increases K⁺ conductance of the vesicles.

We agree with the reviewer about the argument of basal slow permeation of K⁺ across membranes. The titration of valinomycin in the stopped-flow data (Fig. 6G-H in paper 1) suggests that the basal permeation, if any, must be very low. Further, the basal permeation cannot be measured well because of the small signal. When we performed the flux experiments, any basal permeation of K⁺ (> 0.4 fS) in the experimental time period of 10 minutes for replacing extravesicular KCl would have diminished the Cl⁻ gradient and have failed our flux assays. It means that any basal permeation must be tiny in conductance (<< 0.4 fS).

11) pages 13&14, "*Our data show that the CHGB conducts Cl⁻ and F⁻ much better than Br⁻, I⁻, NO₃⁻, SCN⁻, formate or citrate (Fig. 3G), suggesting that the CHGB is much more selective than other known chloride channels or transporters.*" It's not clear whether light scattering as used in the figure is a convincing indicator of selectivity. One can imagine, for example, that the different light scattering is caused, at least partially, by differences in shape/size of the vesicles when different ions are used for vesicle packing.

In the light-scattering-based assay, our signal comes from ~10E11 vesicles. The averaged size of vesicles would follow roughly a Gaussian distribution. The average structural factor of the vesicles therefore is defined by the averaged feature among the large number of vesicles, and is pretty well controlled by the average size and shape of extruded vesicles. The initial shape of the vesicles is spherical. The increase in Stokes radii reflects the change in the average structural factor of the large number of vesicles caused by the shape change. It is therefore pretty good to follow the average behavior of vesicles when we start with such a huge number of them. The statistical consideration and the titration experiments in Fig. 6E/6F of paper 1 made it clear that a change of relative permeation by up to 100 folds can be measured reliably. It is therefore possible to use the flux assay to provide a good estimate of anion selectivity.

12) Page 16, "*As negative controls, vesicles prepared with BSA, CHGA, a CHGBΔMIF and CHGB-MIF all failed to generate any channel activity (data not shown), suggesting that the observed activity was genuine to the CHGB.*" Please give numbers of trials either in the main text or in the method section.

Each experiment was repeated at least three times but most of them 5 times. We also mentioned this in the figure legends.

13) Page 19, "The control siRNAs with scrambled nucleotide sequences (scRNAs) had no effect (top row in Figs 6A & S6E)." Those figures don't seem to have "controls" without transfection; therefore one can't conclude that transfection of control siRNA (scRNA) had no effect.

The figure was so congested so we did not include cells without any treatment in the now Figs 4/5 in paper 2. We tested DND-160 staining and ratiometric measures of secretory granules in the PC-12 cells independently to make sure that pH levels were at ~5.5, which lead to the conclusion of "scRNAs had no effects". The control data are now added to Fig. 5B (grey bar).

As controls for Figs 4/5 in paper 2, cells with scrambled siRNAs would be a much better negative control because we wanted to compare scRNAs with CHGB-specific siRNAs.

14) page 19, "In order to compare the wild-type CHGB and CHGB Δ MIF, we transiently overexpressed them in CHGB-knockdown cells (Fig. 6A, rows 3 & 4)." Are those mouse CHGB constructs resistant to the siRNA designed against native rat CHGB?

Rat and mouse CHGB proteins have very high sequence identity so that the mouse CHGB mRNAs would not be resistant to the rat siRNAs. It is the transiently overexpressed mRNAs from the introduced gene under a strong promoter that superseded the negative effects of siRNAs.

15) Fig. 6, the dye perhaps also loads other acidic organelles such as lysosomes. If so, what are the criteria for organelle selection used in the pH measurement?

pHlourin is a pH sensitive green fluorescent protein mutant. Syncollin is a granular protein. Syncollin has sorting signal for its granular delivery. Cells expressing Syncollin-pHlourin were stained with DND-160 for imaging. The individual imaging channels for DND-160 (yellow) and pHlourin (green) were recorded and compared. The merged image shows nearly perfect overlapping of them (Fig. S2A in paper 2), suggesting that almost all DND-160-stained acidic compartments are secretory granules with Syncollin. This is not surprising because it is well known that most intracellular acidic compartments in INS-1 cells are secretory granules. Further, the same was already established in literature by other researchers such as Steiner et al. 2006.

• Patrick Stiernet, Yves Guiot, Patrick Gilon, and Jean-Claude Henquin (2006) Glucose Acutely Decreases pH of Secretory Granules in Mouse Pancreatic Islets MECHANISMS AND INFLUENCE ON INSULIN SECRETION THE JOURNAL OF BIOLOGICAL CHEMISTRY VOL. 281, NO. 31, pp. 22142–22151.

Referee #3:

P.5. „A tightly membrane-associated form of CHGB shows up on the surface of PC-12 cells after stimulated granule release". Since PC-12 cells have been intensely studied by patch-clamping in the last decades, are there any reports about a channel with properties the authors found in reconstitution systems?

We cannot find much published data about the channel properties of the CHGB protein in PC-12 membrane. The reported 250 pS Cl⁻ channel reported by Hordejuk et al. from chromaffin granules might be the CHGB channel because the difference in conductance from 125 pS of CHGB channel in 150 mM KCl may likely result from the increased conductance with high concentration of KCl (500 /150 mM KCl by Hordejuk et al.

Confirmation of the molecular identity of this 250 pS native channel will need a lot of different assays. We foresee the needs of highly specific inhibitor, genetic manipulation of the native channel in chromaffin cells, and a biochemical confirmation to eliminate a probability of trace amount of contamination in biochemical preparation of chromaffin granules. These will need to be reported as a separate study.

Reference:

Hordejuk, R., et al. (2006). "The heterogeneity of ion channels in chromaffin granule membranes." *Cell Mol Biol Lett* 11(3): 312-325.

p.12 : Channels able to conduct F⁻ are usually assumed to be also permeable to OH⁻ ions. Since the anion channel in secretory vesicles is involved in pH setting of these organelles such a function might be physiologically important and should be addressed by the authors.

This could not be measured directly. Our prediction from data in Fig. 4D in paper 1 is that the OH⁻ flux is negligible. Because the concentration of OH⁻ is about six orders of magnitude lower than Cl⁻ in cytosol, the conduction of OH⁻ is negligible in comparison to Cl⁻ flux.

p. 13: The sentence that „CHGB is much more selective than other known anion channels" is misleading. What the authors want to say is that the channel prefers small anions like Cl⁻ and F⁻ over large ones, in contrast to the lyotropic selectivity sequence of many other channels such as CaCC, VRAC or CFTR. Could the authors provide a number for the anion/cation permeability ratio (chloride/sodium or chloride potassium?)

We do not have data for the anion/cation ratio because the measurements in bi-ionic conditions in the lipid bilayers were not reliable for cations due to the small number of channels in each patch.

p.15: The time axis in the recording in Fig. 5A is not correct. I assume it is sec instead of ms.

The “x 10³” mark in the axial label in the last version was not easy to recognize, although the label was correct. We relabeled the figure in revision.

p.15: In the supplemental text, the authors report that channel activity is unusually only sustained for a couple of minutes. This information should be provided and discussed in the main text. Moreover, given its role in pH regulation, modulation of channel activity by pH should be studied and shown.

The stability of the bilayer is not equivalent to the stability of channel in membrane. Even though CHGB proteins do not cause membrane cracks, they may oligomerize and remodel the bilayer membrane as showed in Fig. 2 of paper 2. Both pH and Ca²⁺ modulations of the CHGB channel will be interesting to study, especially to identify specific residues that are responsible for such effects. We will perform such studies in detail for a separate publication. With the current data, our channel recordings at pH 5.5 (Fig. 4 in paper 1) and our flux assays at pH7.4 (Fig. 5A in paper 1) show that the CHGB channel is functional in both pH7.4 and 5.5.

Discussion: A high conductance anion channel will clamp the vesicle potential to the Nernst potential for Cl⁻. Hence, high intravesicular [Cl⁻] will increase the electrochemical gradient for V-type ATPases and impairs acidification. The authors should discuss potential consequences of this electrical signal.

Because the Cl⁻ flux follows the pumping of H⁺, it is a passive process and the net result is nearly complete neutralization of the accumulated positive charges by H⁺-ATPase. The positive potential inside granules will help maintain a high concentration gradient of Cl⁻ ions across granule membranes.

Discussion: Because of its particular biophysical properties, CIC-3 is expected to accumulate chloride in vesicles and thus to impair acidification by depolarizing the synaptic vesicle. In light of this reasoning, the phenotypic similarity of Clcn3^{-/-} and CHGB KO animals is intriguing, especially as the question whether CIC-3 is present in granules is not yet settled. The authors should address the aspect of the CIC- 3-CHGB interaction to the discussion.

Because of different views and results in literature on CIC-3, we concentrate the discussions of CIC-3^{-/-} and CHGB KO to the DISCUSSION section in paper 1. Specific genetic manipulations will be needed in a future study in order to directly test the possible CLC-3 and CHGB interactions.

Referee #4:

I am not an expert in cryo-EM, but I am skeptical that one can obtain a reliable 10Å resolution structure from only 6900 particles. The authors need to be more rigorous in explaining how this structure was solved.

The referee's concern is not valid any more with advanced cryoEM and high-quality data from direct electron detectors. These days it is very common to obtain near-atomic resolution structures with ~10,000 particles. The data we collected at Janelia Farm were from a Falcon 2 detector without movie function and were not as good as more advanced direct electron detectors, contributing to the limited resolution for our map calculated from 6,900 particle images.

Here is one example.

Alan Merk, Alberto Bartesaghi, Soojay Banerjee, Veronica Falconieri, Prashant Rao, Mindy I. Davis, Rajan Pragani, Matthew B. Boxer, Lesley A. Earl, Jacqueline L.S. Milne and Sriram Subramaniam (2016). Cell 165, 1698–1707. (For the final reconstruction of glutamate dehydrogenase, only ~21800 particles are used and resolution obtained was 1.8Å).

The authors conclude that the structure that migrates as ~300kd in SEC is a dimer because it is composed of a 100kd detergent micelle plus two 86kd monomers = 272kd. But if it were a trimer, it would be 358kd. It is not at all clear that the resolution of their SEC could distinguish between 272kd and 358kd.

Yes, that is true. We could not distinguish between 272kDa and 385kDa. But our size-exclusion data were backed up by the cryoEM data analysis and 3D model (Figs 1C & 1D in paper 1). Multivariate statistical analysis of the images revealed strong C2 symmetry. The reconstructions showed C2 even without imposition of symmetry (Fig. 1D, S2H in paper 1).

Helix 3, which is the presumptive transmembrane helix, has a significant proportion of hydrophilic amino acids. But, the authors seem to take their data at face value and do not dig deep enough. A few questions that come to mind that the authors might have investigated include: what is the energy required to insert this into a lipid bilayer? If the amino acids in this helix are scrambled, does it still float with lipid? Is the MIF also resistant to digestion by other proteases?

These are all good directions for future studies. We prepared a recombinant protein containing the MIF, and found that it interacts avidly with membrane (Fig. 3E in paper 1). It would be interesting to obtain the structure of this protein in membrane by solid-state NMR. The reviewer's questions will be addressed in a separate study via systematical analysis of the MIF-membrane interaction by perturbing the helix 3 hydrophobicity.

The single channel currents in Fig. 5 are uninterpretable. What is happening in Fig. 5A? What are the controls?

Negative controls included vesicles without CHGB and vesicles with CHGB-MIF, CHGBΔMIF and CHGA, none of which showed any channel activity after fusion into bilayers.

Why do the authors use the electrophysiological method only in Fig. 3B and then switch to the light scattering assay? I am not sure that I believe the validity of the light scattering assay. Are there other anion channels that conduct both F and Cl? It is hard to understand how F and Cl are conducted, but Br is not. Dissecting the mechanism of conduction alone would be a significant contribution, but here it is buried.

We switched to the flux assay because the bilayer recordings from dozens or hundreds of CHGB channels were not feasible due to membrane breakdown observed when more CHGB channels were fused, which probably reflects the membrane-remodeling property of CHGB (Fig. 2F in paper 1). Further, the flux assay allowed us to test billions of channels at the same time, eliminating the possible activity from trace amount of contaminating channels.

The Fluc family of F- channels are selective for F- and Cl- although they are much more conductive for F- over Cl-. Below are the references. The flux assay we used was developed by Stockbridge et al in a paper published in eLife (2013). To enhance the understanding of the CHGB channel, we generated a deletion of half of the loop between helices 2 & 3 (CHGB Δ L1), and found that the deletion mutant showed significant impairment in anion flux (Fig. 5E in paper 1) when compared with wild-type protein.

References for the Fluc channel:

Turman, Daniel L; Nathanson, Jacob T; Stockbridge, Randy B et al. (2015) Two-sided block of a dual- topology F- channel. Proc Natl Acad Sci USA 112:5697-701.

Ji, Chunhui; Stockbridge, Randy B; Miller, Christopher(2014)Bacterial fluoride resistance, Fluc channels, and the weak acid accumulation effect. J Gen Physiol 144:257-61.

The discussion of the membrane potential of the vesicles in item 22 of the supplement is gibberish. Likewise for the figure legend. For example, the statement "With zero extravesicular KCl, the potential ΔV is $\sim +295$ mV" is hard to understand. This depends on the selectivity of the channels that are open. In any case, if there is zero extracellular KCl, the Nernst potential would be infinite. The statement "With the efflux of KCl, the potential would quickly change from $+295$ mV to < 180 mV." I do not understand where the 180 mV comes from.

The discussion is now reorganized in the item 13 in the supplementary information of paper 1. We have now explicitly discussed how a small efflux of Cl- caused the change of the transmembrane electrostatic potential from infinity to $\sim +295$ mV. Given the high selectivity of CHGB channel, this estimate might very likely be within the right range. We also showed in Fig. 5H and 5I in paper 1 that a low concentration of extravesicular Cl- is sufficient to change the electrostatic potential in the vesicles and block almost all efflux signal. The 180 mV was from the reversal potential with 0.2 mM Cl- outside the vesicles and 300 mM KCl inside, which is now labeled in Fig. 5H.

This paper needs to be split into three or more papers and needs to be presented in a clear and concise way.

Because of the unconventional Cl- channel formed by CHGB, we would like to put together all the evidence we have collected. With more data since last submission, we have split the manuscript into two companion papers. The first one is focused on the CHGB channel property in membranes, and the second is on the CHGB channel function in granule acidification and maturation in endocrine cells. The current presentation is much more clear and concise.

Referee #1

Report for Author:

This is a much improved version of an earlier manuscript that included some very interesting results showing that CHGB protein forms a Cl- channel. In this revision, the authors removed several loose conclusions and now focus on the biochemical and biophysical characterization of the protein. Specifically, the authors added trypsin digestion protection experiments and membrane curvature measurements (Fig. 2) to show that purified CHGB can be inserted into membrane bilayers under certain conditions. In addition, the authors added data to the macroscopic Cl- assays using wild-type and mutant CHGB proteins reconstituted in vesicles (Fig. 5E). The data is now fairly convincing that purified CHGB protein can exist as a "soluble" form and as a membrane-inserted form, with the later forming a Cl- channel. It's not clear from the studies whether under physiological conditions, such a Cl- channel can be formed inside a cell, but the authors did a good job in their discussion to address such an uncertainty.

I have only a few minor comments to this revision.

Lines 240-241, "The detected channel is thus Cl⁻ selective with no detectable K⁺ conduction", the authors should calculate PK/PCI with the reversal potentials already available, instead of just saying "no detectable K⁺ conductance".

Lines 241-244, "As negative controls, vesicles prepared with BSA, CHGA (Fig. S4I), a CHGB deletion mutant lacking the CHGB-MIF (CHGB Δ MIF) and CHGB243 MIF itself all failed to generate any channel activity (data not shown), suggesting that the observed channel activity is probably genuine to CHGB." The authors should give numbers to compare between wild-type and the "controls" on how many out of how many preps had currents. Lipid bilayers are notoriously known to generate artifacts and it's prudent to have enough numbers from both the WT and the controls.

Line 226, "The channels were almost always open in a low voltage range (-50 to +50 mV)." The authors should have a more detailed single channel analysis with the voltage-dependency of Po.

Transfer Review:

Yes, my report and identity can be transferred confidentially

Referee #2

Report for Author:

The authors provide an impressive number of approaches to make a strong point that CHGB is an anion channel in secretory vesicles. I find that the division of the paper in two impaired the readability, however, the authors are right in that reduction in length was necessary.

I am disappointed how the authors reacted on my suggestions. Almost every response declares that this cannot be done or will be reported in an additional paper. The authors could have recorded currents in BLM at the concentrations used by Hordejuk et al. (first comment) or measured the anion/cation permeability (third comment). However, this does not change the fact that this manuscript reports important new insights in a convincing manner that will provoke new experiments and greatly contribute to solving an important biological question.

I therefore recommend acceptance of both papers.

Referee #3

Report for Author:

Yaday et al.: Secretory granule protein chromogranin B forms an anion channel in membrane
The authors have now subdivided their data into two papers. I review the first paper. My main problem is still with the measurements of single channels in planar lipid bilayers Fig. 4., which I find unpublishable. When performing such experiments, it is necessary to show unequivocally, in each experiment, that these are currents through ion-selective ion channels, as opposed to currents through unspecific transient lipid channels - this is even more important when introducing a protein found to introduce lipid modeling. However, the channels in Fig. 4A are obtained under symmetrical conditions (150 mM chloride on both sides), so the reversal potential of both chloride selective and unselective channels will be at 0 mV. In Fig. 4C-D, the currents are only measured in one direction, and the reversal potential is found by extrapolation. This is not enough, and it is ludicrous to use this loose extrapolation to state in the Abstract that CHGB displays "a higher anion selectivity than other five known families of Cl⁻ channels" (if that is the basis for the statement). It is not acceptable to answer this important point by referring to the difficulty of detecting channel opening when choosing to only have 1.5 mM Chloride on the cis side: the authors could have done other experiments! What about choosing 150 and 30 mM chloride? The experiments in Fig. 4E-4F again have a reversal potential very close to zero. Overall, the authors have not established that they are measuring a current through a selective chloride channel. This could be transient channels formed through lipid under the influence of the protein. These experiments must be repeated under properly controlled conditions or taken out. Whether Fig. 5 delivers enough data on a chloride permeability without Fig. 4, I can't judge.

Other points:

1. p. 10, line 194: "secondary structure analysis". How was this done?
 2. p. 9: the authors conclude that almost all CHGB was put into vesicles from the outside, because it is readily digested by trypsin. This also makes sense because a superficial membrane protein inserted into one side of a membrane will cause the tubulation the authors show in Fig. 2. Such assays are very unspecific - it basically just means the protein is inserted into one leaflet of the membrane. However, how does this agree with the suggestion that CHGB forms a transmembrane ion channel, which is packed into the vesicle from the inside in the TGN?
 3. The authors should show original data (like in Fig. 4A) showing progressive DIDS-blocking, not only the overall curve in Fig. 5B.
 4. Like mentioned by another reviewer in the last round, it is not clear that the conclusion that CHGB conducts Fluoride and chloride better than the other anions can be drawn from the data in Fig. 4. for the other anions in panel F: what is the control experiments, showing that the vesicles can be deformed under these conditions (when formed with Bromide, for instance)?
 5. Where are the controls for experiments in Fig. 4E showing that different mutations do not cause large increase in scattering? how is it established that they reconstituted in vesicles to the same degree as WT protein?
 6. the experiment in Fig. 5G has not been sufficiently described.
 7. Why does the section "Random reconstitution and a Poisson consideration suggest that the CHGB channel is a tetramer", end up with speculation about valinomycin? It seems out of place for this manuscript.
 8. The point 13 in the supplementary is opaque to me. What does "After a small leak of Cl⁻ to the outside, an electrostatic potential was established. The Nernst equation was related to the charging of the vesicle after ~580 chloride ions moved outside" mean? Why 580 chloride ions? A solution to what problem was obtained in MATLAB?
- Minor points:
9. p. 5: "CHGB has thus dual states - outside or inside membranes." This makes no sense. What does it mean?
 10. p. 6, line 103: Fig. 1C. Do the authors mean Fig. 1B?
 11. Why is the gel in Fig. 2E also presented in Fig. S3D?
 12. line 962: "We followed closely what was described in the paper and discussed with Dr. Stockbridge" This seems to be some kind of appeal to authority, which is not appropriate. The authors should describe what they did. They can thank Dr. Stockbridge in the Acknowledgements, if they want to, but do not make Dr. Stockbridge indirectly responsible for this paper.
 13. The control trace in Fig. S4H looks very strange - there is a lot of very repetitive noise on the baseline, which appears to be 'cut' somehow. What happened here?
 14. line 281-282: Cl⁻/H⁺ co-transporter. Do the authors mean anti-porter? Otherwise, it would be an electroneutral transporter?
 15. Line 303-304: "Relative flux of F⁻ and Cl⁻ is consistent with the measured permeation ratio". The flux of F⁻ and Cl⁻ was not measured in this assay.

Thank you for transferring your revised manuscript entitled "Secretory granule protein chromogranin B forms an anion channel in membrane" to Life Science Alliance. Your manuscript was reviewed twice at a different journal before, and the journal editors transferred the previous reviewer reports to us.

Your work proposes that Chromogranin B forms an anion conductance channel in the membrane and is thus the enigmatic chloride channel ensuring anion conductance critical for neuroendocrine vesicle release. The reviewers think your findings are potentially important and offer avenues for future research. They think, however, that alternative explanations for the observed data still exist. Despite this criticism, we would be happy to publish your work in Life Science Alliance if you down-tone your conclusion on anion channel formation in the membrane to allow for alternative explanations (eg transient channels formed through lipid under the influence of the protein as mentioned by reviewer #3). Especially the conclusions drawn from figure 4 need to be clearly dampened in the text, and the title needs to be changed. Furthermore, the minor/other points of reviewer #1 and reviewer #3 should get addressed. In principle, we think that your data could get further strengthened if using a changed protocol of single channel recording (eg, changing 1.5 mM Chloride on the cis side to 150 and 30mM Chloride and measuring single channel currents at more

potentials to analyse the reversal potential of the channel), so you may want to consider including such an experiment as well.

2nd Revision – authors' response

29 August 2018

During the revision, we added data from bilayer recordings of CHGB channel in asymmetrical Cl⁻ conditions (Fig. 4A) with 42 mM in cis and 166 mM in trans, which set a Nernst potential of Cl⁻ (E_{Cl}) at +34.8 mV. We measured the reversal potentials from either single channel currents vs. V_m (Fig. 4B) or from average macroscopic currents vs. V_m (Fig. S5C), and found that both measurements were ~30 mV, close to the E_{Cl}. The measured single channel conduction in the asymmetrical Cl⁻ conditions was close to the one measured with symmetrical Cl⁻ (Fig. 4C/D). We therefore concluded that the CHGB vesicles indeed gave rise to Cl⁻ selective anion channels, which agreed well with the other electrical recordings and with the direct recordings (Fig. 3C) using the Ag/AgCl electrode, and with the flux assays in Fig. 5. The selectivity of the CHGB channels and the voltage-dependent open probability (red trace in Fig. S5C) made it very different from the published nonselective lipid ion channels. We have added in the discussion section the remote possibility of the lipid-channels formed in the presence of CHGB. As we discussed before, when CHGB in membranes made of DOPC : Sphingomyelin : cholesterol = 3:1:1 we observed a non-selective leak conductance of ~225 pS, which could be easily separated from the smaller conductance of CHGB channel. This point is now mentioned in the text.

The concern raised by review #3 on the selectivity or transient lipid ion channel is now well addressed.

Other points:

1. p. 10, line 194: "secondary structure analysis". How was this done?

In an online server as detailed in the supplementary information. We now added the information to the figure legend.

2. p. 9: the authors conclude that almost all CHGB was put into vesicles from the outside, because it is readily digested by trypsin. This also makes sense because a superficial membrane protein inserted into one side of a membrane will cause the tubulation the authors show in Fig. 2. Such assays are very unspecific - it basically just means the protein is inserted into one leaflet of the membrane. However, how does this agree with the suggestion that CHGB forms a transmembrane ion channel, which is packed into the vesicle from the inside in the TGN?

There is a misunderstanding here. A protein that is membrane protected does not allow the conclusion that the proteins are in one leaflet because the proteins could be inserted into two leaflets. There are multiple examples. Human C-type lectin (RegIIIγ) is a good example, which was used for the experiment in Fig. 3A.

3. The authors should show original data (like in Fig. 4A) showing progressive DIDS-blocking, not only the overall curve in Fig. 5B.
Done as suggested in Fig. 5B and 5I.
4. Like mentioned by another reviewer in the last round, it is not clear that the conclusion that CHGB conducts Fluoride and chloride better than the other anions can be drawn from the data in Fig. 4. For the other anions in panel F: what is the control experiments, showing that the vesicles can be deformed under these conditions (when formed with Bromide, for instance)?

In the flux assays, only F⁻ and Cl⁻ were found to give sufficient signals, as in Fig. 5F/5G. The control experiments is the comparison between different ions, say the nonpermeable isethionate. Based on the theory for light scattering, signals from light scattering measure the average structural

factors of the vesicles, which in our assay are mainly determined by the average Stokes radii of the vesicles. Higher scattering is resulted from larger Stokes radii. Because the solutions were isotonic and did not cause vesicle fusion, the larger average Stokes radius had to come from the ellipsoid or oblong-shapes of the vesicles. In addition, as showed in Fig. 3C, upon addition of valinomycin, the vesicles released Cl⁻, and there ought to be K⁺ efflux in accompany of Cl⁻ release.

5. Where are the controls for experiments in Fig. 4E showing that different mutations do not cause large increase in scattering? how is it established that they reconstituted in vesicles to the same degree as WT protein?

It was Fig. 5E, not 4E. We pointed out this detail in the supplementary information. For all flux assays, we always in parallel reconstituted wild-type CHGB as positive control and used empty vesicles as negative control. The signals in Fig. 5E were normalized. For comparison, the same amounts of proteins were used in reconstitution. The relative amount of protein inserted in membranes was assayed by running the gel after vesicle floatation- assay, which was performed in the same way as described in Fig. 3G and 3H.

6. the experiment in Fig. 5G has not been sufficiently described.

We added more explanation in supplementary info. Instead of using K-isethionate in the outside of the vesicles, KBr, KI, and KF were used to replace KCl before the light-scattering based flux assays.

7. Why does the section "Random reconstitution and a Poisson consideration suggest that the CHGB channel is a tetramer", end up with speculation about valinomycin? It seems out of place for this manuscript.

This is something that came out of the titration. We think that it is a tightly related point for the flux assays.

8. The point 13 in the supplementary is opaque to me. What does "After a small leak of Cl⁻ to the outside, an electrostatic potential was established. The Nernst equation was related to the charging of the chloride ions? A solution to what

problem was obtained in MATLAB?

A small number of Cl⁻ ions came out through the CHGB channel in order to establish a steady state positive potential which prevents further net efflux of Cl⁻. The number of Cl⁻ ions (W) that came out vesicles to reach a steady-state was calculated based on the charging of the vesicles by such amounts of ions, assuming a Cole membrane capacitance coefficient of 1 uF/cm², and an average vesicle size of 100 nm. The Nernst equation and the charging of the vesicles both depend on W. An analytical solution of W was not derived. Instead, we obtained a numerical solution in MATLAB. We have added the equation for W and described the details on finding the W.

Minor points:

9. p. 5: "CHGB has thus dual states - outside or inside membranes." This makes no sense. What does it mean?

It means two different physical states of the protein. There are multiple other proteins that exist as a soluble form and as an integrated protein in membranes. Melittin, C-type lectin, alpha-hemolysin are good examples. We have discussed this point sufficiently in the text.

10. p. 6, line103: Fig. 1C. Do the authors mean Fig. 1B? Changed.

11. Why is the gel in Fig. 2E also presented in Fig. S3D? Removed from Fig. S3D.

12. line 962: "We followed closely what was described in the paper and discussed with Dr. Stockbridge" This seems to be some kind of appeal to authority, which is not appropriate. The

authors should describe what they did. They can thank Dr. Stockbridge in the Acknowledgements, if they want to, but do not make Dr. Stockbridge indirectly responsible for this paper.

This is common sense. We have removed it so that Dr. Stockbridge would not be responsible for what we did in the paper. We added her name to the acknowledgements.

13. The control trace in Fig. S4H looks very strange - there is a lot of very repetitive noise on the baseline, which appears to be 'cut' somehow. What happened here?

There was some noise from the fluorometer due to stirring bar. It was not a cut of the signal. We changed to a different experimental trace in Fig. 4H.

14. line 281-282: Cl-/H+ co-transporter. Do the authors mean anti-porter? Otherwise, it would be an electroneutral transporter?

There seems confusion in terminology from the reviewer. A co-transporter is a broader name. It can be symporter or anti-porter, and can be electroneutral or electrogenic. A co-transporter does not have to be an electroneutral transporter. Under this nomenclature, the Cl-/H+ co-transporter is an electrogenic antiporter.

15. Line 303-304: "Relative flux of F- and Cl- is consistent with the measured permeation ratio". The flux of F- and Cl- was not measured in this assay.

Not sure what the reviewer really meant. The flux assay reflected an indirect measurement of the anion flux that caused the sudden change of vesicle Stokes radii. The titration we performed in Fig. 6 suggested that it is possible to measure the relative flux because the flux did vary among vesicles have zero, one, two or more channels.

2nd Editorial Decision

11 September 2018

Thank you for submitting your Research Article entitled "Secretory granule protein chromogranin B forms an anion channel in membrane". I discussed your revised work with an advisor again, who appreciates the introduced changes and supports publication (see comment below). It is thus a pleasure to let you know that your manuscript is now accepted for publication in Life Science Alliance. Congratulations on this interesting work.

Comment of the advisor:

The authors have addressed all reviewers' concerns. I suggest acceptance of the manuscript.